# Probing the role of synaptic adhesion molecule RTN4RL2 in setting up cochlear connectivity

Nare Karagulyan[1,2,3†], Maja Überegger[4†], Yumeng Qi[5†], Norbert Babai[1,2], Florian Hofer[4], Lejo Johnson Chacko[6], Fangfang Wang[5], Maria Luque[6], Rudolf Glueckert[6], Anneliese Schrott-Fischer[6], Yunfeng Hua[5]*, Tobias Moser[1,2,3,7]*, Christine Bandtlow[4]*

[1]Institute for Auditory Neuroscience, University Medical Center Göttingen, Göttingen, Germany; [2]Auditory Neuroscience and Synaptic Nanophysiology Group, Max Planck Institute for Multidisciplinary Sciences, Göttingen, Germany; [3]Cluster of Excellence "Multiscale Bioimaging: From Molecular Machines to Networks of Excitable Cells", Göttingen, Germany; [4]Institute of Neurobiochemistry, Biocenter, Medical University of Innsbruck, Innsbruck, Austria; [5]Shanghai Institute of Precision Medicine, Ninth People's Hospital, Shanghai Jiao Tong University School of Medicine, Shanghai, China; [6]Department of Otorhinolaryngology, Medical University of Innsbruck, Innsbruck, Austria; [7]Department for Hearing, Speech and Voice Disorders, Medical University of Innsbruck, Innsbruck, Austria

*For correspondence:
yunfeng.hua@shsmu.edu.cn (YH);
tmoser@gwdg.de (TM);
christine.bandtlow@i-med.ac.at (CB)

†These authors contributed equally to this work

Competing interest: The authors declare that no competing interests exist.

## eLife Assessment

In this work, the authors characterize the synaptic adhesion molecule RTN4RL2, demonstrating its critical involvement in the development and function of auditory synapses between inner hair cells and spiral ganglion neurons. This study is **important** because it offers potential insights into therapeutic strategies for hearing loss associated with synaptic dysfunction. The findings are **solid**, because they are supported by the use of multiple advanced techniques, including FISH and SBEM imaging.

**Abstract** Sound encoding depends on the precise and reliable neurotransmission at the afferent synapses between the sensory inner hair cells (IHCs) and spiral ganglion neurons (SGNs). The molecular mechanisms contributing to the formation, as well as interplay between the pre- and postsynaptic components, remain largely unclear. Here, we tested the role of the synaptic adhesion molecule and Nogo/RTN4 receptor homolog RTN4RL2 (also referred to as NgR2) in the development and function of afferent IHC–SGN synapses. Upon deletion of RTN4RL2 in mice (RTN4RL2 KO), presynaptic IHC active zones showed enlarged synaptic ribbons and a depolarized shift in the activation of $Ca_V1.3$ $Ca^{2+}$ channels. The postsynaptic densities (PSDs) of SGNs were smaller and deficient of GluA2–4 AMPA receptor subunits despite maintained *Gria2* mRNA expression in SGNs. Next to synaptically engaged PSDs, we observed 'orphan' PSDs located away from IHCs, likely belonging to a subset of SGN peripheral neurites that do not contact the IHCs in RTN4RL2 KO cochleae, as found by volume electron microscopy reconstruction of SGN neurites. Auditory brainstem responses of RTN4RL2 KO mice showed increased sound thresholds indicating impaired hearing. Together, these findings suggest that RTN4RL2 contributes to the proper formation and function of auditory afferent synapses and is critical for normal hearing.

## Introduction

Hearing relies upon the correct formation and maturation of cochlear afferent synapses between postsynaptic type I spiral ganglion neurons (SGNs) and presynaptic inner hair cells (IHCs; reviewed in *Johnson et al., 2019*; *Bulankina and Moser, 2012*; *Appler and Goodrich, 2011*). In mice, SGN neurites extend and reach IHCs at late embryonic stages (E16.5; *Koundakjian et al., 2007*), whereby early synaptic contacts are established between the two cells at E18 (*Michanski et al., 2019*). Developmental changes and maturation of afferent synapses continue until the onset of hearing (p12), with further refinement occurring till the fourth postnatal week (*Wong et al., 2014*; *Liberman and Liberman, 2016*; *Michanski et al., 2019*). In mature cochleae, each IHC receives a contact from 5 to 30 SGNs, while each SGN contacts only one active zone (AZ) of one IHC in the majority of cases (*Meyer and Moser, 2010*; *Hua et al., 2021*).

To this date, multiple mechanisms have been implicated in SGN neurite guidance and establishment of IHC innervation by SGNs, including signaling via EphrinA5–EphA4, Neuropilin2/Semaphorin3F, Semaphorin5B/PlexinA1, and Semaphorin3A (*Defourny et al., 2013*; *Coate et al., 2015*; *Jung et al., 2019*; *Cantu-Guerra et al., 2023*). Less is known about the molecules governing the transsynaptic organization at IHC afferent synapses. Similar to the central conventional synapses, synaptic adhesion proteins such as neurexins and neuroligins have been suggested to play a role in pre- and postsynaptic assemblies (*Ramirez et al., 2022*; *Jukic et al., 2024*). Furthermore, the $Ca_V1.3$ extracellular auxiliary subunit $Ca_V\alpha_2\delta_2$ was indicated to be important for proper alignment of presynaptic IHC AZs and postsynaptic densities (PSDs) of SGNs (*Fell et al., 2016*).

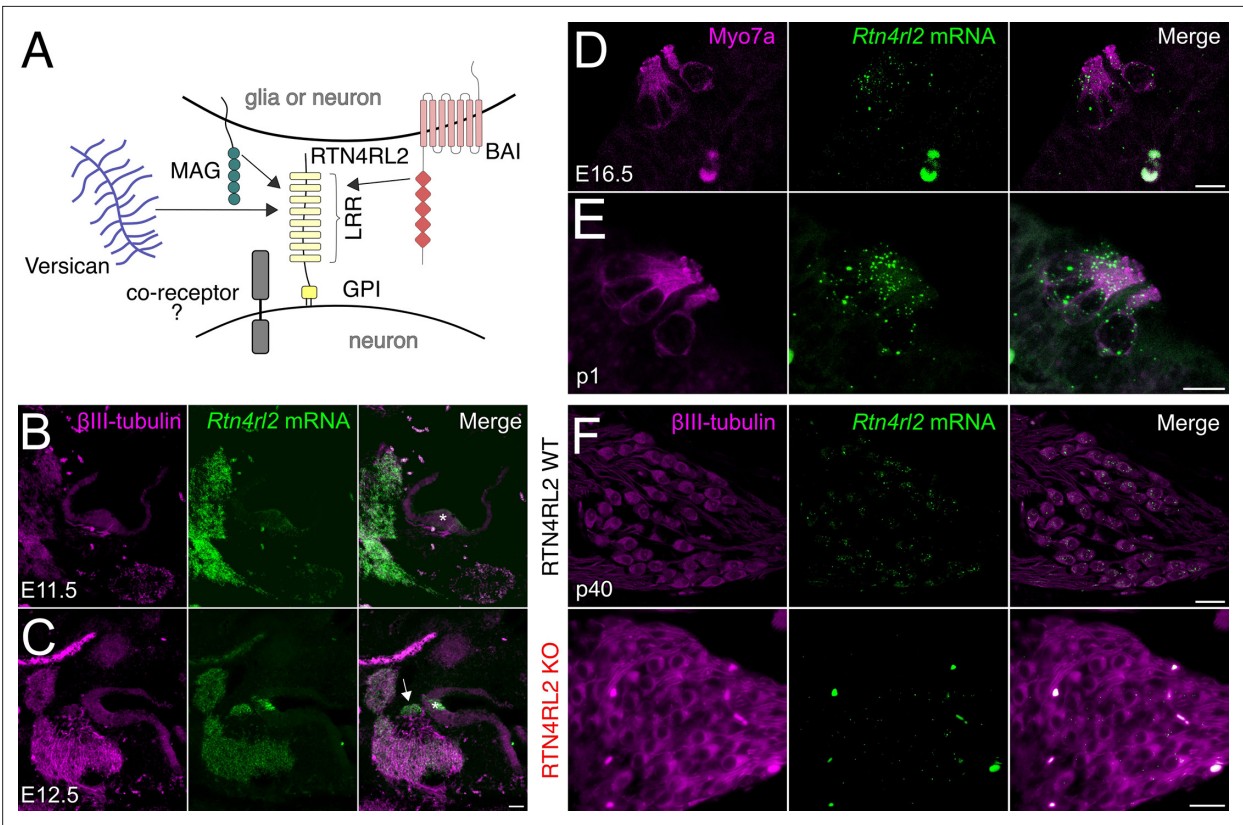

**Figure 1.** RTN4RL2 mRNA expression in inner hair cells (IHCs) and spiral ganglion neurons (SGNs) of the mouse cochlea. (**A**) RTN4RL2 is an LRR protein and is anchored to the cell membrane via glycosylphosphatidylinositol (GPI). In the nervous system, RTN4RL2 has been implicated to interact with MAG, versican, brain-specific angiogenesis inhibitor (BAI) (*Venkatesh et al., 2005*; *Bäumer et al., 2014*; *Wang et al., 2021*). Exemplary images of RNAscope ISH for *Rtn4rl2* mRNA (green) in the otic vesicles of E11.5 (**B**) and E12.5 (**C**) mice. Scale bar = 50 µm. The developing organ of Corti region is marked with an asterisk and the spiral ganglion is indicated with an arrow. Exemplary images of *Rtn4rl2* mRNA expression in hair cells of E16.5 (**D**) and p1 (**E**) mice. Hair cells are visualized with anti-Myo7a stainings. Scale bars = 10 µm. (**F**) Representative images of RNAscope ISH for *Rtn4rl2* mRNA (green dots) combined with immunostaining for neuron-specific marker βIII-tubulin (gray) in paraffin sections of p40 RTN4RL2 WT and RTN4RL2 KO cochleae. Scale bars = 20 µm.

Whether reticulon 4 receptors (RTN4Rs) contribute to setting up afferent connectivity in the cochlea remained to be investigated. The RTN4 receptor family consists of three homologous proteins RTN4R, RTN4RL1, and RTN4RL2. RTN4Rs are leucine-rich repeat (LRR) and glycosylphosphatidylinositol anchored cell surface receptors (*Figure 1A*). Their primary role is thought to limit synaptic plasticity and axonal outgrowth as well as to restrict axonal regeneration after injury (reviewed in *Mironova and Giger, 2013*). While RTN4R and RTN4RL1 are involved in axonal guidance (*Vaccaro et al., 2022*), RTN4RL2 has been proposed to be important for innervation of the epidermis by dorsal root ganglion neurons (*Bäumer et al., 2014*). Furthermore, RTN4Rs are suggested to control the number and the development of synapses (*Wills et al., 2012*) and to play a role in transsynaptic signaling (*Wang et al., 2021*).

Recent single-cell transcriptomic studies of the cochlea detected the expression of RTN4Rs in SGNs and IHCs (*Shrestha et al., 2018*; *Jean et al., 2023*). Here, we investigated the role of RTN4RL2 in the cochlea using previously described RTN4RL2 constitutive knock-out mice (RTN4RL2 KO; *Wörter et al., 2009*). We found RTN4RL2 to be expressed in SGNs and to be required for normal hearing: auditory brainstem responses (ABRs) were impaired upon RTN4RL2 deletion. We discovered both pre- and postsynaptic alterations of IHC–SGN synapses in RTN4RL2 KO mice: presynaptic Ca$^{2+}$ channels of IHCs required stronger depolarization to activate and PSDs seemed deficient of GluA2–4 AMPA receptor subunits. Additionally, a subset of type I SGN neurites did not contact IHCs but likely still feature 'orphan' PSDs.

## Results

### RTN4RL2 is expressed both in hair cells and SGNs

Recent transcriptomic analyses have revealed *Rtn4rl2* expression both in hair cells and SGNs (*Elkon et al., 2015*; *Liu et al., 2018*; *Shrestha et al., 2018*; *Jean et al., 2023*). We verified this by performing RNAscope staining in cochleae during inner ear development. We observed *Rtn4rl2* mRNA expression at E11.5 in the hair cell region and at E12.5 in both the spiral ganglion and hair cells (*Figure 1B, C*). *Rtn4rl2* expression was further evident in hair cells at E16.5 and p1 (*Figure 1D, E*) and was maintained at low levels in spiral ganglia of p40 mice (*Figure 1F*). RTN4RL2 KO mice were lacking *Rtn4rl2*-specific mRNA puncta (*Figure 1F*), demonstrating specific detection of *Rtn4rl2* expression in SGNs.

### RTN4RL2 is important for the correct development of the auditory afferent synapses

To probe the role of RTN4RL2 in the cochlea, we first studied the numbers of cochlear cells in RTN4RL2 KO mice. We did not observe any change in SGN density or counts of inner and outer hair cells at p15, 1-month-, and 2-month-old mice (*Figure 2—figure supplement 1*). Given that RTN4RL2 has been implicated in synapse formation and development (*Wills et al., 2012*; *Borrie et al., 2014*; *Wang et al., 2021*), we immunolabeled IHC afferent synapses for presynaptic RIBEYE/Ctbp2 and postsynaptic Homer1 in mice at the age of 3 weeks. While we did not detect any change in the number of synaptic ribbons in RTN4RL2 KO IHCs, yet, ribbon volumes were bigger (*Figure 2C, D*; *Figure 2—figure supplement 2*). Interestingly, in addition to the Homer1-positive puncta juxtaposing presynaptic ribbons, we observed additional Homer1 patches, which appeared to be away from IHCs, potentially marking 'orphan' PSDs (*Figure 2A, B*). Moreover, the Homer1 puncta juxtaposing IHC AZs were significantly smaller in RTN4RL2 KO mice compared to the control IHCs (*Figure 2E*). While we found the percentage of presynaptic ribbons juxtaposing Homer1 immunofluorescent puncta to be decreased by approximately 7% in RTN4RL2 KO mice (*Figure 2F*), we cannot exclude that our immunolabeling protocol lacked the sensitivity to detect smaller synaptically engaged PSDs. Out of the four pore-forming AMPA receptor subunits (GluA1–4), mature SGNs express GluA2–4 (*Niedzielski and Wenthold, 1995*; *Matsubara et al., 1996*). Our immunolabeling of anti-GluA2/3 revealed a severe reduction in the number of GluA2/3-positive puncta at RTN4RL2 KO IHCs, indicating possible decrease or lack of GluA2/3 on the postsynaptic side of SGNs (*Figure 3A, B*). Interestingly, the remaining GluA2/3 patches did not properly juxtapose presynaptic ribbons (*Figure 3C*). Despite this, the expression of *Gria2* mRNA appeared to be maintained in SGNs, as indicated by RNAscope (*Figure 3D*). We then checked whether the potential lack of GluA2/3 was accompanied by an increase in GluA4 signal at the PSDs of RTN4RL2 KO SGNs, as was previously shown in GluA3 KO mice (*Rutherford*

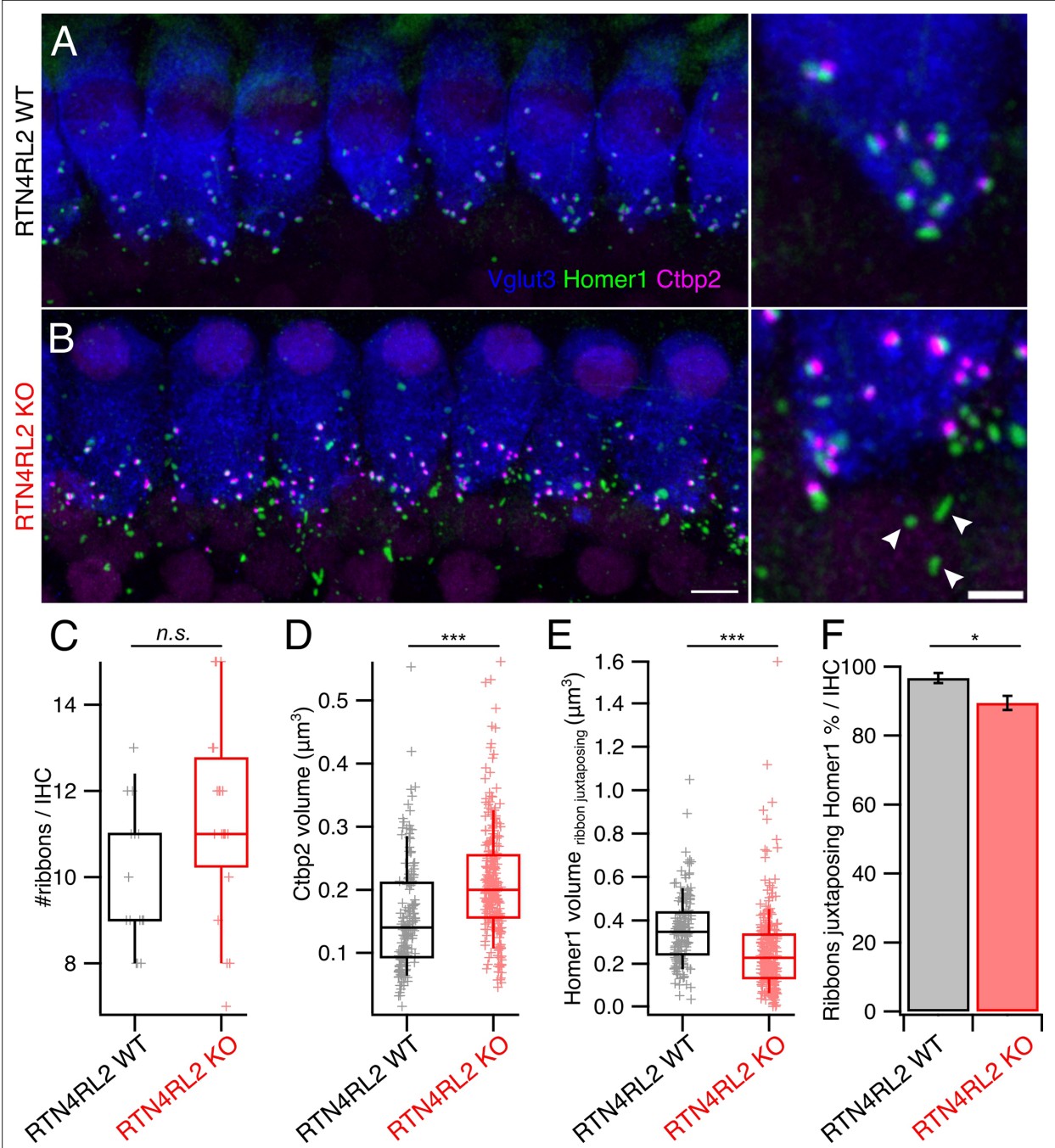

**Figure 2.** Pre- and postsynaptic changes at inner hair cells (IHCs) of RTN4RL2 KO mice. Maximum intensity projections of representative confocal stacks of IHCs from apical cochlear region of 3-week-old RTN4RL2 WT (**A**) and RTN4RL2 KO (**B**) mice immunolabeled against Vglut3, Homer1, and Ctbp2. Scale bar = 5 μm. Images on the right-hand side are zoomed into the synaptic regions. Scale bar = 2 μm. Some of the putative 'orphan' postsynaptic densities (PSDs) are marked with the white arrowheads. (**C**) Number of Ctbp2-positive puncta is not changed in IHCs of RTN4RL2 KO mice (RTN4RL2 WT: 9.9 ± 0.42, SD = 1.62, $n$ = 15, $N$ = 2 vs RTN4RL2 KO: 11.4 ± 0.5, SD = 2.25, $n$ = 20, $N$ = 3; p = 0.06, Mann–Whitney–Wilcoxon test). (**D**) Ribbon volumes are enlarged in RTN4RL2 KO IHCs (RTN4RL2 WT: 0.16 ± 0.007 μm³, SD = 0.09 μm³, $n$ = 165, $N$ = 2 vs RTN4RL2 KO: 0.21 ± 0.005 μm³, SD = 0.09 μm³, $n$ = 259, $N$ = 3; p < 0.001, Mann–Whitney–Wilcoxon test). (**E**) Homer1 patches which are juxtaposing presynaptic ribbons show decreased volumes in RTN4RL2 KO IHCs (RTN4RL2 WT: 0.36 ± 0.01 μm³, SD = 0.16 μm³, $n$ = 160, $N$ = 2 vs RTN4RL2 KO: 0.26 ± 0.01 μm³, SD = 0.19 μm³, $n$ = 249, $N$ = 3; p < 0.001, Mann–Whitney–Wilcoxon test). (**F**) Percentage of Ctbp2 puncta juxtaposing Homer1 is slightly decreased in RTN4RL2 KO mice (RTN4RL2 WT: 96.7 ± 1.45%, SD = 5.44 %, $n$ = 14, $N$ = 2 vs RTN4RL2 KO: 89.5 ± 2.05%, SD = 9.18 %, $n$ = 20, $N$ = 3; p = 0.03, Mann–Whitney–Wilcoxon test). Data is presented as mean ± SEM. Box–whisker plots show the median, 25/75 percentiles (box), and 10/90 percentiles (whiskers). Individual data points are overlaid. Significances are reported as *p < 0.05 and ***p < 0.001.

*Figure 2 continued on next page*

*Figure 2 continued*

The online version of this article includes the following source data and figure supplement(s) for figure 2:

**Source data 1.** Numerical data of *Figure 2C–F*.

**Figure supplement 1.** Cochlear cell densities are not changed in RTN4RL2 KO mice.

**Figure supplement 2.** Intact number but enlarged size of the ribbons in RTN4RL2 KO inner hair cells (IHCs).

**Figure supplement 3.** Efferent innervation pattern in RTN4RL2 KO cochleae.

*et al., 2023*). Surprisingly, we did not detect a significant GluA4 signal compared to the background in RTN4RL2 KO cochleae (*Figure 3—figure supplement 1*). To check if the 'orphan' PSDs away from IHCs were possibly erroneously engaged with efferent presynaptic terminals, we stained the latter for synapsin, which is lacking from IHCs, but did not observe any obvious juxtaposition between the 'orphan' Homer1 and synapsin puncta (*Figure 2—figure supplement 3*).

## Additional non-synaptic neurites in the cochlea of RTN4RL2 KO mice

To further examine the afferent cochlear connectivity, we utilized serial block-face scanning electron microscopy (SBEM) to 3D reconstruct the apical cochlea segments from two RTN4RL2 KO and one littermate RTN4RL2 WT control mice at p36 (*Figure 4A*). As recently demonstrated (*Hua et al., 2021*; *Lu et al., 2024*), the spatial resolution of SBEM allows reliable ribbon synapse identification and neurite reconstruction (*Figure 4B, C*). In these three datasets, we have traced all neural fibers from the habenula perforata (HP; three in each dataset). This resulted in 133 fibers, of which 115 were classified as peripheral neurites of type I SGN based on their radial calibers and extensive myelination after entering the HP (*Figure 4D*). We observed a substantial number of neurites that did not engage the IHCs (non-synaptic neurites) in the RTN4RL2 KO organs of Corti (23 out of 83) in contrast to the wild-type control (3 out of 32, *Figure 4E*). Although most non-synaptic neurites were found in the inner spiral bundle, they failed to reach IHC basal lateral poles to form contacts (17 out of 23 in RTN4RL2 KO animals). This result might provide a plausible explanation for the 'orphan' Homer1 puncta shown by immunohistochemistry in RTN4RL2 KO organs of Corti. Note that the presence of non-synaptic neurites in the RTN4RL2 KO animals was not associated with a profound reduction in ribbon synapses (*Figure 2C*, *Figure 2—figure supplement 2B*, *Figure 4—figure supplement 1A*), suggesting a different scenario than deafferentation due to excitotoxic synaptopathy as recently reported (*Moverman et al., 2023*). Nevertheless, nearly all traced radial fibers were found predominantly unbranched in both the RTN4RL2 KO (97%) and the wild-type control mice (94%; *Figure 4F*).

## Deletion of RTN4RL2 results in depolarized shift of the Ca²⁺ channel activation in IHCs

Next, we tested for potential effects of RTN4RL2 deletion on the presynaptic IHC function. We performed whole-cell patch-clamp recordings from IHCs of p21-29 RTN4RL2 WT and RTN4RL2 KO mice. First, we recorded voltage-gated $Ca^{2+}$ currents by applying step depolarizations with 5 mV increment in ruptured patch-clamp configuration (*Figure 5A*, see Materials and methods). We did not observe any change in the maximal $Ca^{2+}$ current amplitude (*Figure 5B, C*), yet noticed small (~+3 mV) but consistent and statistically significant depolarized shift of the voltage of half-maximal $Ca^{2+}$ channel activation ($V_{half}$; *Figure 5D, E*). The voltage sensitivity of the channels was not changed in RTN4RL2 KO IHCs (*Figure 5F*). Next, we probed $Ca^{2+}$ influx triggered exocytosis from IHCs by applying depolarizations of varying durations to voltages saturating $Ca^{2+}$ influx in IHCs of both genotypes (–17 mV) and recording exocytic membrane capacitance changes ($\triangle C_m$) in the perforated patch-clamp configuration (*Figure 5G*). Both fast (up to 20 ms depolarization) and sustained components of exocytosis remained intact in the mutant, indicating unaffected readily releasable pool and replenishment of the vesicles in RTN4RL2 KO IHCs (*Figure 5H*).

We then turned to study single AZ function by combining whole-cell ruptured patch-clamp with spinning disc confocal imaging of presynaptic $Ca^{2+}$ influx, as described previously (*Ohn et al., 2016*). TAMRA-conjugated Ctbp2 binding dimeric peptide and the low affinity $Ca^{2+}$ indicator Fluo4-FF ($k_D$: 10 μM) were loaded into the cell via the patch pipette and $Ca^{2+}$ influx at single AZs was visualized in the form of hotspots by applying voltage ramp depolarizations to the IHCs and simultaneously imaging Fluo4-FF fluorescence near labeled ribbons (*Figure 6*, see Materials and methods). Maximal

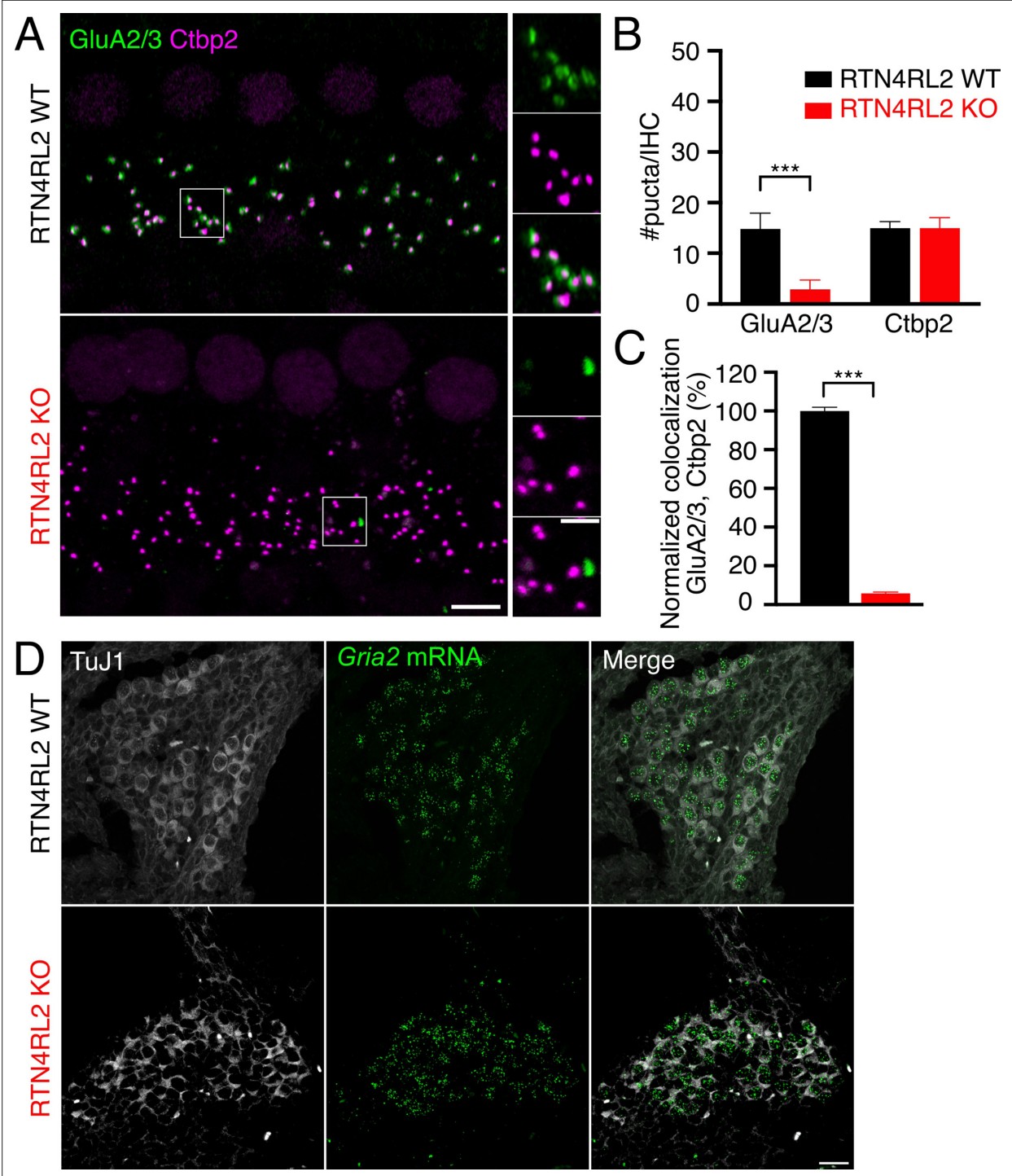

**Figure 3.** Reduced GluA2/3 signal juxtaposing presynaptic ribbons in inner hair cells (IHCs) of RTN4RL2 KO mice. (**A**) Maximum intensity projections of representative IHC regions from 1- to 1.5-month-old RTN4RL2 WT (top) and RTN4RL2 KO (bottom) mouse cochleae immunolabeled against Ctbp2/Ribeye (ribbons) and GluA2/3 (AMPA receptors of postsynaptic density [PSD]). Scale bar = 5 µm. The zoom-in regions marked with the white rectangles are presented on the right-hand side. Scale bar = 2 µm. (**B**) The number of the GluA2/3-positive puncta is drastically reduced in RTN4RL2 KO mice despite the maintained number of presynaptic ribbons (p < 0.001, Mann–Whitney–Wilcoxon test). (**C**) Disrupted colocalization of Ctbp2/Ribeye and GluA2 immunofluorescence puncta at IHCs of RTN4RL2 KO mice (p < 0.001, Mann–Whitney–Wilcoxon test). *N* = 6 animals/genotype. (**D**) Representative images of RNAscope ISH from p4 mice show maintained expression of Gria2 (red dots) in the spiral ganglion neuron (SGN) somata of RTN4RL2 KO mice. Scale bar = 20 µm. Significances are reported as ***p < 0.001.

The online version of this article includes the following source data and figure supplement(s) for figure 3:

*Figure 3 continued on next page*

*Figure 3 continued*

**Figure supplement 1.** No apparent GluA4 signal juxtaposing presynaptic ribbons at inner hair cell (IHC) synapses of RTN4RL2 KO mice.

**Figure supplement 1—source data 1.** Numerical data of *Figure 3—figure supplement 1D*.

$Ca^{2+}$ influx at single AZs tended to be higher in IHCs of RTN4RL2 KO mice, but the difference did not reach statistical significance (*Figure 6B, B'*). The $V_{half}$ of $Ca^{2+}$ channel clusters at single AZs showed depolarized shift ($\sim$+5 mV), which was statistically significant despite the large variability of $V_{half}$ across different AZs (*Figure 6C, C'*). Furthermore, we did not observe any mismatch in Ctbp2-positive puncta and evoked $Ca^{2+}$ hotspots, indicating the correct localization of $Ca^{2+}$ channels at the AZs. We verified this by performing immunostainings of $Ca_V 1.3$, Ctbp2, and Bassoon and did not detect any apparent misalignment of these protein clusters (*Figure 6D*). In summary, deletion of RTN4RL2 causes a depolarized shift of the activation of presynaptic $Ca^{2+}$ influx in IHCs as demonstrated at the single AZ and whole-cell levels. This is expected to elevate the sound pressure levels required to reach a comparable activity of synaptic transmission for sound encoding and predicts elevated auditory thresholds, which we tested by recording ABRs.

## RTN4RL2 is essential for normal hearing

In order to evaluate auditory system function in RTN4RL2 KO mice, we recorded ABRs at the age of 2–4 months. ABRs were elicited in response to 4, 8, 16, and 32 kHz tone bursts and click stimuli of increasing sound pressure levels. We determined the sound thresholds by identifying the lowest sound pressure level which resulted in detectable ABR waveform. We found ABR thresholds to be significantly increased by around 30–45 dB across all the tested frequencies and in response to the click stimulations in RTN4RL2 KO mice compared to the wild-type controls (*Figure 7*). RTN4RL2 heterozygous (RTN4RL2 HET) mice showed an intermediate phenotype between the RTN4RL2 KO and RTN4RL2 WT mice with a lower yet significant increase of ABR thresholds of approximately 10–15 dB at 4, 16, and 32 kHz frequencies.

## Discussion

In this study, we investigated the role of the Nogo/RTN4 receptor homolog RTN4RL2 in the cochlea. Consistent with the previous reports, we detected RTN4RL2 expression in IHCs and SGNs. Upon RTN4RL2 deletion, we observed alterations of the IHC–SGN synapses (*Figure 8*). Presynaptically, $Ca^{2+}$ channel activation was shifted toward depolarized voltages, and ribbon size was enlarged. IHC exocytosis was unaltered when probed at saturating depolarizations. At the postsynaptic side, PSDs juxtaposing presynaptic ribbons were smaller compared to the controls and seemed to be deficient of GluA2–4, despite the expression of *Gria2* mRNA in RTN4RL2 KO SGNs. Additionally, we observed PSDs which resided further from the IHC membrane and did not juxtapose the presynaptic AZs in RTN4RL2 KO mice. These PSDs potentially belong to the type I SGN neurites that ended in the inner spiral bundle without contacting the IHCs, as observed in SBEM reconstructions of RTN4RL2 KO SGNs. Finally, ABR thresholds were elevated in RTN4RL2 KO mice, indicating that RTN4RL2 is required for normal hearing.

RTN4 receptors have been implicated as presynaptic adhesion molecules based on their transsynaptic interactions with postsynaptically enriched brain-specific angiogenesis inhibitor (BAI) adhesion GPCRs (*Stephenson et al., 2013*; *Wang et al., 2021*). Despite this, at least RTN4R has been proposed to function postsynaptically, given that its knockdown leads primarily to postsynaptic changes (*Wills et al., 2012*). The interpretation of RTN4R function and localization is further complicated by their enrichment in both pre- and postsynaptic terminals (*Wang et al., 2002a*). Similarly, we observed RTN4RL2 expression in both IHCs and SGNs. Interestingly, a recent study described a hearing impairment in BAI1-deficient mice and suggested that postsynaptically functioning BAI1 is important for correct localization of AMPA receptor subunits in SGNs based on the absence or decrease of different AMPA receptor subunits at the SGN postsynapses in BAI1-deficient mice (*Carlton et al., 2024*). Similarly, our data from RTN4RL2 KO mice show a major reduction of GluA2–4-positive puncta juxtaposing presynaptic AZs. Importantly, the BAI1 knock-out model in the study by Carlton et al. eliminated specifically long isoform of BAI1, which contains extracellular thrombospondin repeats (TSRs) important for mediating transsynaptic interactions, while leaving the short isoform – containing the transmembrane

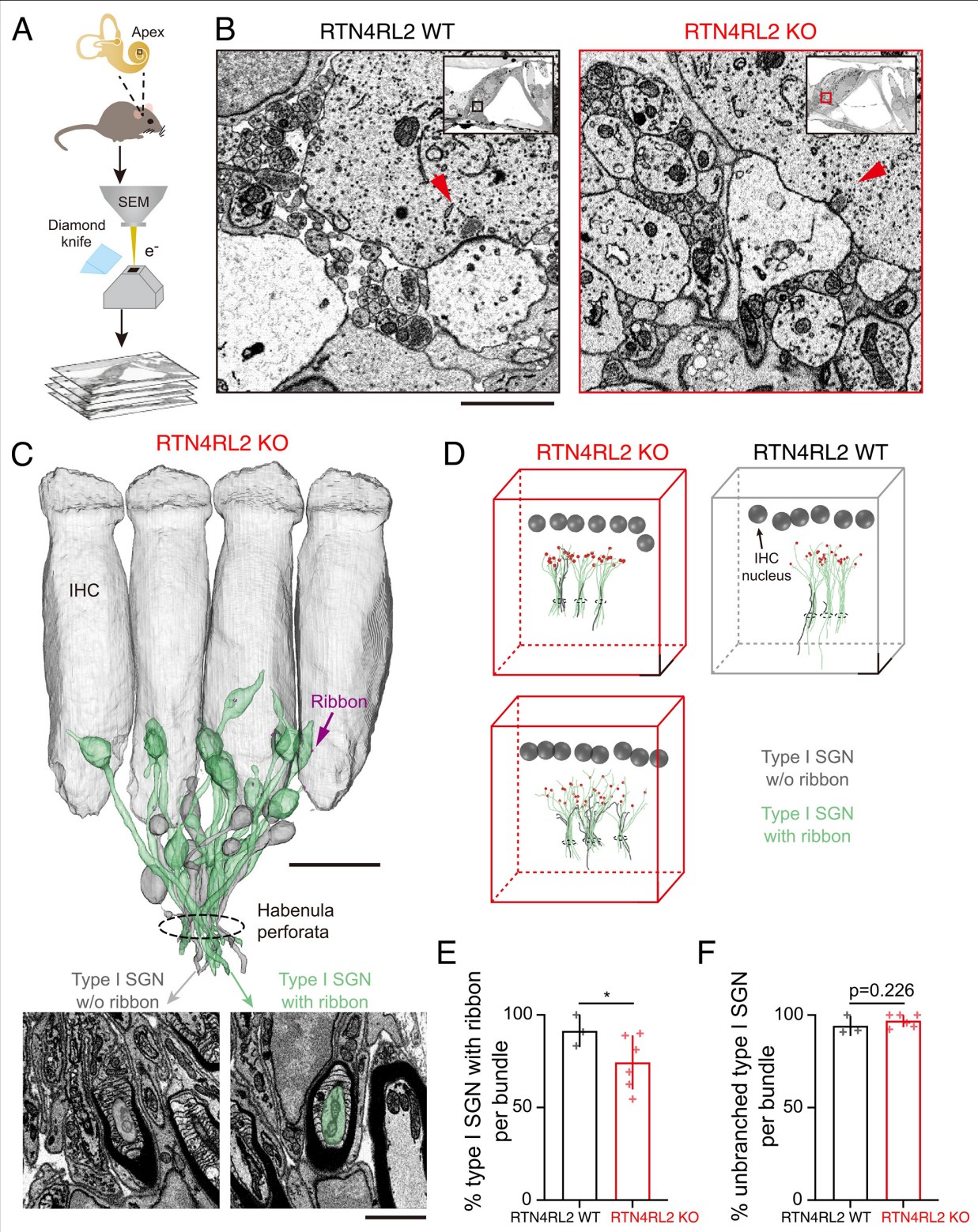

**Figure 4.** Additional non-synaptically engaged spiral ganglion neuron (SGN) neurites in the cochlea of RTN4RL2 KO mice. (**A**) Workflow of serial block-face scanning electron microscopy (SBEM) imaging at the mouse apical cochlear region. (**B**) Example images of neurites beneath inner hair cells (IHCs) from the RTN4RL2 WT (left) and RTN4RL2 KO (right) mice. Synaptic ribbons are indicated with red arrows. The regions of interest were magnified from single sections of SBEM datasets (insets). Scale bar = 2 μm. (**C**) 3D rendering of afferent fiber reconstruction with ribbons (red), showing both synaptic (green, type I SGN with ribbon) and non-synaptic (gray, type I SGN without ribbon) populations in the RTN4RL2 KO mouse. Scale bar = 10 μm. Inset: EM

*Figure 4 continued on next page*

*Figure 4 continued*

images show the SGN segments on which myelin sheaths start to form. Scale bar 2 = μm. (**D**) Display of classified radial fibers in the RTN4RL2 WT (right) and RTN4RL2 KO (left and bottom) animals. All fibers were traced from the habenula perforata (cycles) before classification to avoid bias to terminal types. Scale bar = 10 μm. (**E**) Percentage of radial fibers with ribbon per bundle (RTN4RL2 WT: 91.41 ± 8.35%, $n$ = 3 bundles, $N$ = 1 vs RTN4RL2 KO: 74.40 ± 14.58%, $n$ = 6 bundles, $N$ = 2; $p$ = 0.032, unpaired $t$-test). (**F**) Percentage of unbranched radial fibers per bundle (RTN4RL2 WT: 94.19 ± 5.04%, $n$ = 3 bundles, $N$ = 1 vs RTN4RL2 KO: 96.98 ± 3.49%, $n$ = 6 bundles, $N$ = 2; $p$ = 0.226, unpaired $t$-test). Significances are reported as *$p$ < 0.05.

The online version of this article includes the following source data and figure supplement(s) for figure 4:

**Source data 1.** Numerical data of *Figure 4E, F*.

**Figure supplement 1.** Quantification of ribbon number and volume in serial block-face scanning electron microscopy (SBEM) reconstructions.

**Figure supplement 1—source data 1.** Numerical data of *Figure 4—figure supplement 1A, B*.

repeats and intracellular domain of the receptor – intact (*Wang et al., 2021*; *Carlton et al., 2024*). Along with the evidence for transsynaptic interaction between RTN4Rs and BAIs (*Wang et al., 2021*), this raises the possibility that the GluA2–4 AMPA receptor subunit localization to SGN PSDs requires the interaction of presynaptic RTN4RL2 and postsynaptic BAI1. Likewise, the changes of presynaptic $Ca^{2+}$ channel properties and ribbons in IHCs could result from direct presynaptic action of RTN4RL2. Alternatively, RTN4RL2 could act postsynaptically in SGNs with an impact on transsynaptic signaling to IHCs. Presynaptic changes upon postsynaptic manipulation have been reported previously. For instance, the deletion of postsynaptically expressed Pou4f1 transcription factor resulted in hyperpolarized shift of IHC $Ca^{2+}$ channels, as well as in changes of the presynaptic heterogeneity of IHC AZs (*Sherrill et al., 2019*). Similarly, presynaptic ribbon sizes were altered in IHCs of mice, which exhibited conditional deletion of Runx1 transcription factor from SGNs (*Shrestha et al., 2023*). Further experiments, employing mouse mutants with specific RTN4RL2 deletion in IHCs or SGNs, are needed to evaluate the precise presynaptic and postsynaptic roles of RTN4RL2, including a possible transsynaptic signaling to IHC AZs.

We propose that the afferent synaptic changes contribute to the severe hearing impairment in RTN4RL2 KO mice. The phenotype of increased ABR thresholds could partially be explained by the depolarized activation of $Ca^{2+}$ channels in IHCs. As it has been shown, the apparent $Ca^{2+}$ dependence of glutamate release from IHCs on average follows a near linear relationship (*Wong et al., 2014*; *Özçete and Moser, 2021*; *Jaime Tobón and Moser, 2023a*; *Jaime Tobón and Moser, 2023b*). Therefore, due to the depolarized shift of the $Ca^{2+}$ channel activation, the glutamate release from IHCs would require stronger stimuli. While exocytosis, probed by depolarizing the IHCs to –17 mV and recording cell membrane capacitance changes, remained intact in RTN4RL2 KO mice, it could still be affected at milder depolarizations (e.g., –50 to –40 mV), which correspond to the receptor potential range of the IHCs. This could result in increased sound thresholds, as shown in Ribeye knock-out mice, where depolarized activation of $Ca^{2+}$ channels has been associated with higher firing thresholds and lower spontaneous rates in SGNs, although a less severe hearing phenotype was observed in those mice (*Jean et al., 2018*). Furthermore, we have shown recently that a direct manipulation of the voltage dependence of $Ca_v1.3$ $Ca^{2+}$ channel activation results in changes of SGN firing and potentially auditory thresholds, further supporting our hypothesis of sound threshold elevation in RTN4RL2 KO mice being caused by the depolarized activation of $Ca^{2+}$ channels in IHCs (*Karagulyan et al., 2025*).

In addition or alternatively, the increase in auditory thresholds could be caused by the deficiency of GluA2–4 AMPA receptors of the SGN postsynapses in RTN4RL2 KO mice. Compared to GluA3 KO mice, which display reduced GluA2 signal at SGN PSDs but intact auditory thresholds, the increased auditory thresholds in RTN4RL2 KO mice could additionally be caused by the lack of GluA4 subunits juxtaposing presynaptic ribbons, which, in contrast, were shown to be upregulated in GluA3 KO mice (*Rutherford et al., 2023*). Given the observed synaptic changes, it seems less likely that the hearing impairment would have resulted from the intrinsic changes of SGN firing. Further experiments, where SGNs would be stimulated directly via ex vivo patch-clamp or optogenetically, could help to probe for functional deficits in SGNs of RTN4RL2 KO mice. Another interesting observation was the additional PSDs not juxtaposing presynaptic ribbons that were detected around IHCs of RTN4RL2 KO mice. These 'orphan' PSDs seem not to be part of the efferent synapses formed between LOC/MOCs and SGNs as we verified by simultaneous anti-Homer1 and anti-synapsin immunolabelings. To check the SGN connectivity, we performed SBEM in RTN4RL2 WT and RTN4RL2 KO cochlear samples and

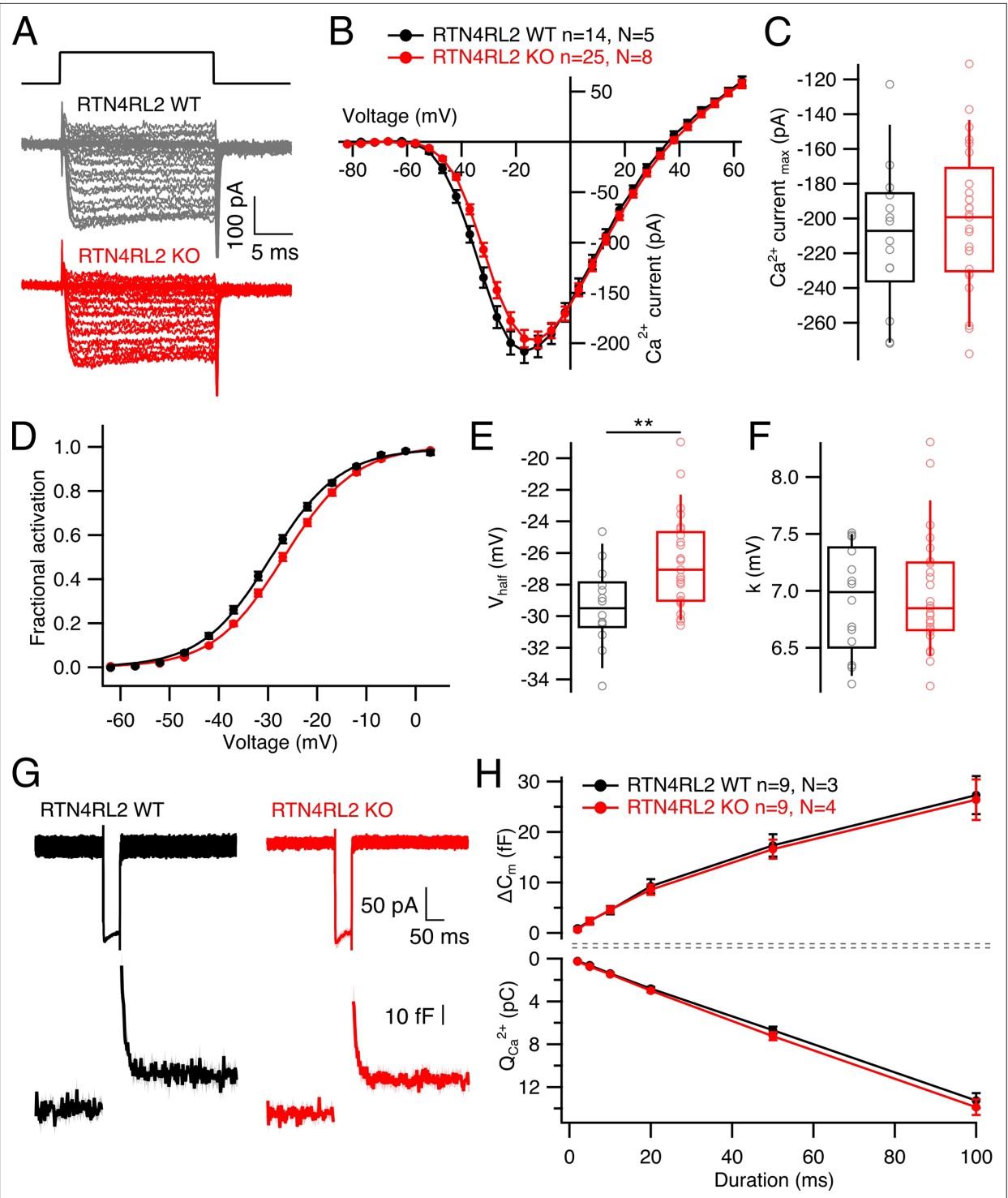

**Figure 5.** Shifted operation range of Ca²⁺ channels but intact exocytosis in inner hair cells (IHCs) of RTN4RL2 KO mice. (**A**) Representative current traces from IHCs of RTN4RL2 WT (top, black) and RTN4RL2 WT (bottom, red) evoked by step depolarizations. (**B**) Average Ca²⁺ current–voltage relationships (*IV* curves) in RTN4RL2 WT and RTN4RL2 KO IHCs. (**C**) Maximal Ca²⁺ current amplitude is not changed in RTN4RL2 KO IHCs (RTN4RL2 WT: 210 ± 10.9 pA, SD = 40.9 pA, *n* = 14, *N* = 5 vs RTN4RL2 KO: –200 ± 8.19 pA, SD = 40.9 pA, *n* = 25, *N* = 8; p = 0.47, Student's *t*-test). (**D**) Fractional activation curves of Ca²⁺ channels calculated from the *IV* curves show depolarized shift in channel activation in RTN4RL2 KO IHCs. (**E**) Voltage of half maximal activation obtained from Boltzmann fit of the curves from (**D**) is more positive in RTN4RL2 KO IHCs (RTN4RL2 WT: –29.4 ± 0.66 mV, SD = 2.48 mV, *n* = 14, *N* = 5 vs RTN4RL2 KO: –26.6 ± 0.59 mV, SD = 2.94 mV, *n* = 25, *N* = 8; p = 0.004, Student's *t*-test). (**F**) Voltage sensitivity (*k*) is not changed in RTN4RL2 KO IHCs (RTN4RL2 WT: 6.92 ± 0.12 mV, SD = 0.46 mV, *n* = 14, *N* = 5 vs RTN4RL2 KO: 6.98 ± 0.1 mV, SD = 0.51 mV, *n* = 25, *N* = 8; p = 0.93, Mann–

*Figure 5 continued on next page*

observed a substantial percentage of SGN neurites in the inner spiral bundles in RTN4RL2 KO samples that were not synaptically engaged with IHCs. They might potentially house the 'orphan' PSDs observed around the IHCs in our immunolabeled samples. How exactly those non-synaptic neurites appear remains to be investigated. SGN neurite retraction from IHCs has been demonstrated in noise-exposed animals, followed by the degeneration of the SGN somata and axons (*Kujawa and Liberman, 2009*; *Kujawa and Liberman, 2015*; *Moverman et al., 2023*). This process, however, results in up

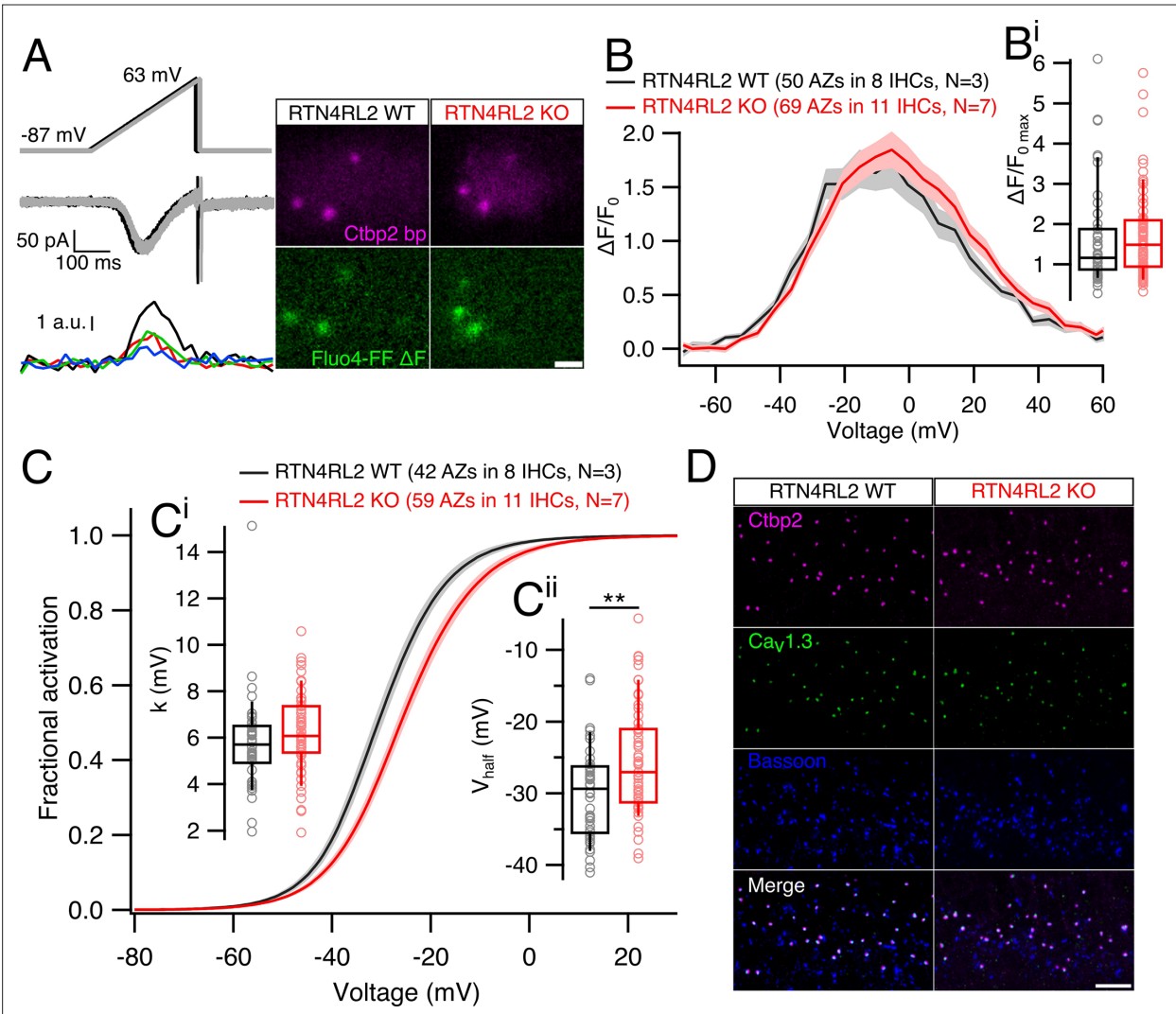

**Figure 6.** Depolarized shift of Ca²⁺ channel activation at single active zones (AZs) but intact presynaptic organization in RTN4RL2 KO inner hair cells (IHCs). (**A**) Voltage ramp stimulation protocols (top), evoked whole-cell currents (middle), and the presynaptic hotspots of Fluo4-FF fluorescence (bottom) of a representative IHC recording. Black and gray colors represent the two stimulations, one being 5 ms shifted over the other. Images on the right show single imaging planes of representative RTN4RL2 WT (left) and RTN4RL2 KO (right) IHCs filled with TAMRA-conjugated Ctbp2 binding peptide (Ctbp2 bp) and Fluo4-FF Ca²⁺ dye. Ca²⁺ hotspots are visualized by subtracting the average of baseline planes from the average of 5 planes during stimulation. Scale bar = 2 μm. (**B**) Average fluorescence–voltage relationships of Ca²⁺ influx at single AZ from RTN4RL2 WT and RTN4RL2 KO IHCs show no difference in the maximal Ca²⁺ amplitude (**B$^i$**; RTN4RL2 WT: 1.6 ± 0.17, SD = 1.2, $n$ = 50 AZs vs RTN4RL2 KO: 1.7 ± 0.13 pA, SD = 1.09, $n$ = 69 AZs; $p$ = 0.24, Mann–Whitney–Wilcoxon test). Shaded areas represent ± SEM. (**C**) Average fractional activation curves of Ca²⁺ channels at single AZs show intact voltage sensitivity (**C$^i$**; RTN4RL2 WT: 5.79 ± 0.32 mV, SD = 2.04 mV, $n$ = 42 AZs vs RTN4RL2 KO: 6.25 ± 0.22 mV, SD = 1.7 mV, $n$ = 59 AZs; $p$ = 0.06, Mann–Whitney–Wilcoxon test) but depolarized shift of $V_{half}$ (**C$^{ii}$**; RTN4RL2 WT: –30 ± 1 mV, SD = 6.5 mV, $n$ = 42 AZs vs RTN4RL2 KO: –25.5 ± 0.98 mV, SD = 7.49 mV, $n$ = 59 AZs; $p$ = 0.002, Student's $t$-test) in RTN4RL2 KO IHCs. Shaded areas represent ± SEM. (**D**) Representative immunolabelings of presynaptic proteins show no apparent mislocalization in RTN4RL2 KO IHCs. Scale bar = 5 μm. Box–whisker plots show the median, 25/75 percentiles (box), and 10/90 percentiles (whiskers). Individual data points are overlaid. Significances are reported as **$p < 0.01$.

The online version of this article includes the following source data for figure 6:

**Source data 1.** Numerical data of *Figure 6B, C, Bi–Cii*.

*Figure 5 continued*

Whitney–Wilcoxon test). (**G**) Average current traces evoked by 50 ms depolarization to –17 mV (top row) and resulting capacitance response (bottom row) from RTN4RL2 WT (left, black) and RTN4RL2 KO (right, red) IHCs. Shaded areas represent ± SEM. (**H**) Exocytic capacitance change ($\Delta C_m$, top) and corresponding $Ca^{2+}$ charge ($Q_{Ca}^{2+}$, bottom) evoked by depolarizations (to –17 mV) of various durations (2, 5, 10, 20, 50, and 100 ms). Box–whisker plots show the median, 25/75 percentiles (box), and 10/90 percentiles (whiskers). Individual data points are overlaid. Significances are reported as **$p < 0.01$.

The online version of this article includes the following source data for figure 5:

**Source data 1.** Numerical data of *Figure 5B–F, H*.

to 50% degeneration of the ribbon synapses. Interestingly, in RTN4RL2 KO mice, the number of the ribbon synapses as well as the density of the SGN somata was largely intact. This indicates that the non-synaptic neurites do not result from synaptic loss as found in hereditary or acquired synaptopathy (*Roux et al., 2006*; *Ruel et al., 2008*; *Seal et al., 2008*; *Kujawa and Liberman, 2009*). Although it is plausible that the non-synaptic SGN neurites may fail to reach the IHCs during initial development, it is unlikely that this would selectively impact only a subset of neurons and still allow the Homer1-positive patches to develop in the absence of presynaptic components. Alternatively, it is known that major synaptic pruning takes place in developing SGNs, whereby up to 50% of the synapses are lost between the IHCs and the SGNs (*Defourny et al., 2013*; *Coate et al., 2019*), suggesting that the non-synaptic SGN neurites of RTN4RL2 KO mice could be a result of failed pruning of SGNs during the development. Yet how this would then comply with normal synapse and SGN counts remains to be elucidated. Future studies will be required to understand the fate of type I SGNs in RTN4RL2 KO mice during development by mapping their full-length periphery projections with large scale imaging tools.

Future work will be required to identify the RTN4RL2 ligands in the cochlea. To this date, the interaction partners of RTN4Rs are not very well understood. While RTN4R in-trans interactions include NogoA, MAG, OMgp, chondroitin sulfate proteoglycans, the only well-established ligand for RTN4RL2 remains MAG (*Fournier et al., 2001*; *Domeniconi et al., 2002*; *Wang et al., 2002b*; *Venkatesh et al., 2005*; *Schwab, 2010*). Other attractive candidate ligands for RTN4Rs are the BAIs. BAI–RTN4R interaction has been suggested to mediate dendritic arborization, axonal elongation, synapse formation in iPSC-derived neurons (*Wang et al., 2021*). Our study suggests that this interaction might be required for proper GluA2–4 AMPA receptor subunit localization to the SGN PSDs. Finally, RTN4RL2 has been proposed to interact with chondroitin-sulfate proteoglycan versican to

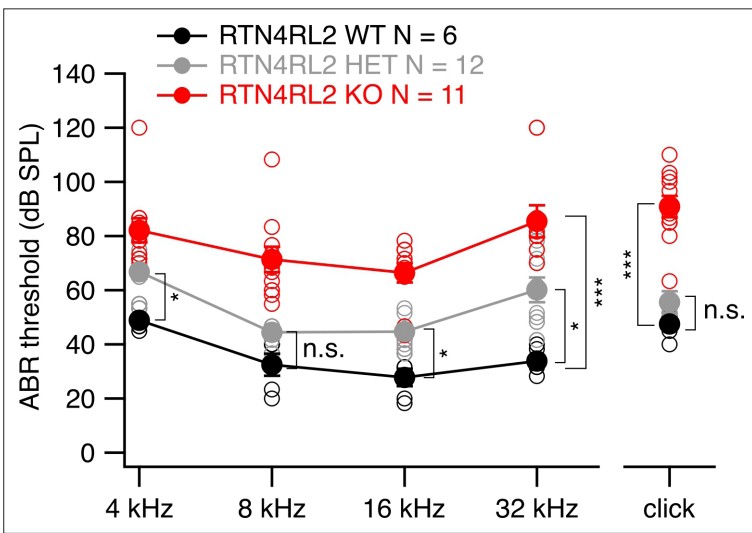

**Figure 7.** Elevated acoustic thresholds in RTN4RL2 KO mice. Auditory brainstem response (ABR) thresholds were measured in response to 4, 8, 16, and 32 kHz tone bursts and click stimuli. ABR thresholds of individual animals are shown in open circles on top of the mean ± SEM. Statistical significances are reported as *$p < 0.05$, ***$p < 0.001$, Kruskal–Wallis followed by Dunn's multiple comparison test.

The online version of this article includes the following source data for figure 7:

**Source data 1.** Numerical data of *Figure 7*.

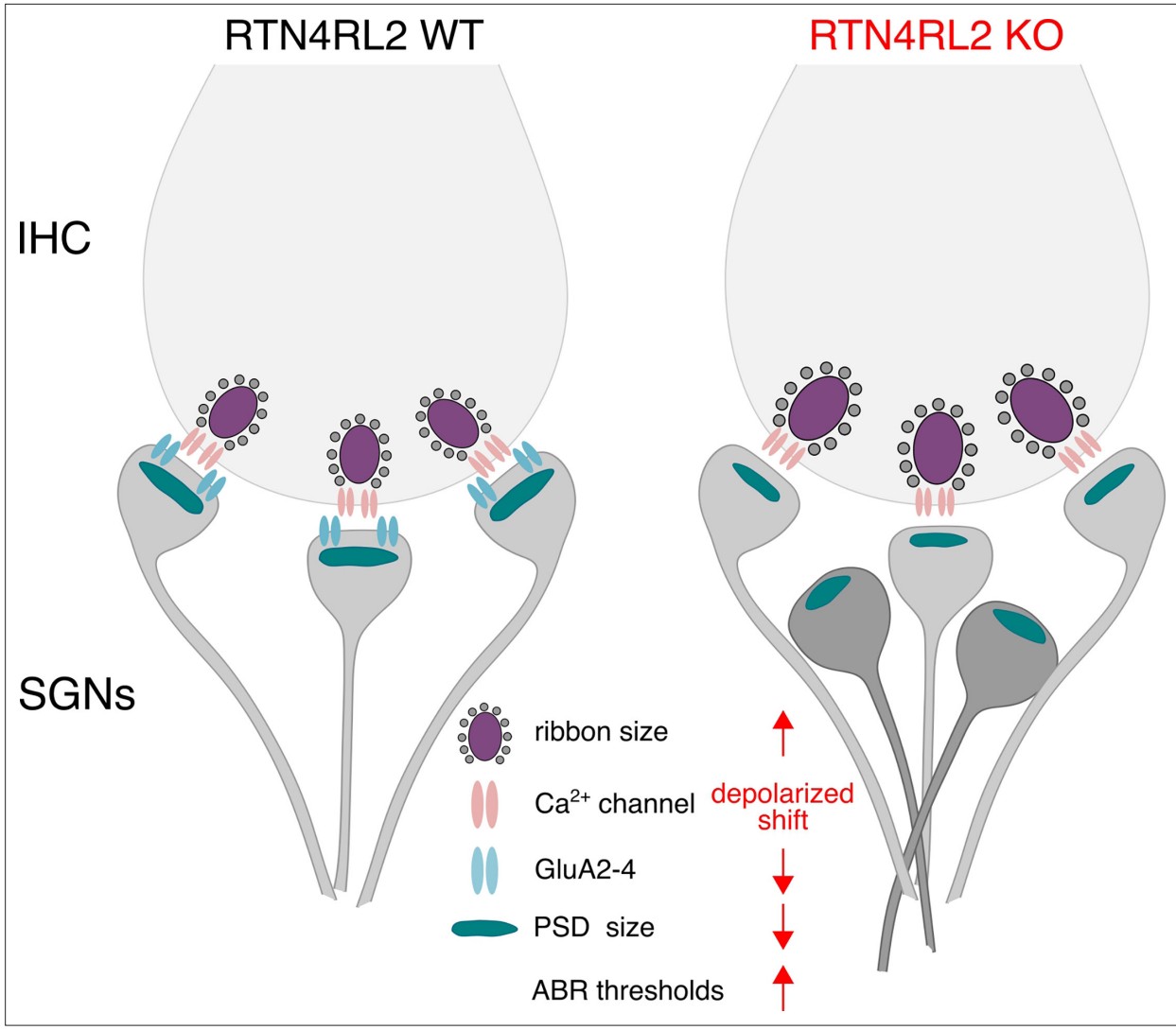

**Figure 8.** Schematic illustration of the key structural and functional changes in the auditory periphery of RTN4RL2 KO mice. RTN4RL2 KO mice display enlarged synaptic ribbons and depolarized shift in the activation of presynaptic $Ca^{2+}$ channels in inner hair cells (IHCs), as well as reduced size of postsynaptic densities (PSDs) juxtaposing presynaptic ribbons. RTN4RL2 deficiency further leads to a decrease in GluA2–4 AMPA receptor subunits at PSDs and results in a subset of type I spiral ganglion neuron (SGN) neurites that reach the inner spiral bundle but do not engage the IHCs.

control the amount of skin innervation by DRG neurites (*Bäumer et al., 2014*). While to our knowledge, attempts have yet to be made for detecting versican signal around the IHCs, other extracellular matrix proteins such as aggrecan, brevican, Tenascin-R, Tenascin-C, hyaluron, and proteoglycan link proteins 1 and 4 have been shown to form dense, basket-like structures that surround the base of the IHCs (*Kwiatkowska et al., 2016*; *Sonntag et al., 2018*). Furthermore, deletion of brevican results in alteration of pre- and postsynaptic spatial coupling at IHC synapses, indicating the possible role of extracellular matrix in transsynaptic organization. However, while the existing observations suggest that the synaptic changes in RTN4RL2-deficient animals might reflect derailed interaction of the SGN neurites with the extracellular matrix, the relevance of such an interaction of RTN4RL2 and versican in the cochlea needs to be addressed in the future.

# Materials and methods

**Key resources table**

| Reagent type (species) or resource | Designation | Source or reference | Identifiers | Additional information |
|---|---|---|---|---|
| Genetic reagent (*Mus musculus*) | RTN4RL2 KO | *Wörter et al., 2009* | | |
| Antibody | anti-Homer1 (rabbit polyclonal) | Synaptic Systems | Cat# 160002 | 1:500 |
| Antibody | anti-Ctbp2 (mouse monoclonal) | BD Biosciences | Cat# 612044 | 1:200 KO validation; https://doi.org/10.7554/eLife.30241 https://doi.org/10.7554/eLife.29275 |
| Antibody | anti-Ca$_V$1.3 (rabbit polyclonal) | Alomone Labs | Cat# ACC-005 | 1:100 KO validation; https://doi.org/10.1523/JNEUROSCI.3411-05.2005 |
| Antibody | anti-Bassoon (mouse monoclonal) | Abcam | Cat# ab8298 | 1:300 |
| Antibody | anti-GluA2 (mouse monoclonal) | Millipore | Cat# MAB397 | 1:200 validated for use in ELISA, IC, IH, IP, RIA, and WB, refer to manufacturer's datasheet |
| Antibody | anti-GluA2/3 (rabbit polyclonal) | Chemicon | Cat# AB1506 | 1:200 validated for use in IH(P), IC, IH, IP, and WB, refer to manufacturer's datasheet |
| Antibody | anti-GluA4 (rabbit polyclonal) | Millipore | Cat# AB1508 | 1:200 KO validation; https://doi.org/10.1126/sciadv.aax5936 |
| Antibody | anti-Myosin7a (rabbit polyclonal) | Abcam/Proteus BioSciences | Cat# ab3481 | 1:800 |
| Antibody | anti-RibeyeA (guinea pig polyclonal) | Synaptic Systems | Cat# 192104 | 1:500 |
| Antibody | anti-Synapsin 1/2 (guinea pig polyclonal) | Synaptic Systems | Cat# 106004 | 1:500 KO validation; refer to manufacturer's datasheet |
| Antibody | anti-Vglut3 (guinea pig polyclonal) | Synaptic Systems | Cat# 135204 | 1:500 KO validation; https://doi.org/10.3389/fncel.2017.00140 |
| Antibody | anti-Parvalbumin (chicken polyclonal) | Synaptic Systems | Cat# 195006 | 1:200 |
| Antibody | anti-Calretinin (chicken polyclonal) | Synaptic Systems | Cat# 214106 | 1:200 |
| Antibody | anti-VaChT (guinea pig polyclonal) | Synaptic Systems | Cat# 139105 | 1:1000 |
| Antibody | anti-ATP1A3 (mouse monoclonal) | Invitrogen | Cat# MA3-915 | 1:300 |
| Antibody | βIII-tubulin (mouse monoclonal) | Biolegend | Cat# 801202 | 1:1000 |
| Antibody | Alexa Fluor 488 conjugated anti-guinea pig (goat polyclonal) | Thermo Fisher Scientific | A11073 | 1:200 |
| Antibody | Alexa Fluor 488 conjugated anti-rabbit (goat polyclonal) | Thermo Fisher Scientific | A11008 | 1:200 |

*Continued on next page*

*Continued*

| Reagent type (species) or resource | Designation | Source or reference | Identifiers | Additional information |
|---|---|---|---|---|
| Antibody | Alexa Fluor 568 conjugated anti-chicken (goat polyclonal) | Abcam | ab175711 | 1:200 |
| Antibody | Alexa Fluor 633 conjugated anti-guinea pig (goat polyclonal) | Thermo Fisher Scientific | A21105 | 1:200 |
| Antibody | STAR 580 conjugated anti-mouse (goat polyclonal) | Abberior | ST580P-1001-500UG | 1:200 |
| Antibody | STAR 635 conjugated anti-rabbit (goat polyclonal) | Abberior | ST635P-1002-500UG | 1:200 |
| Commercial assay, kit | RNAscope | Advanced Cell Diagnostics | | |
| Sequence-based reagent | *Gria2* probe (mouse) RNAscope | Advanced Cell Diagnostics, | Cat# 865091 | |
| Sequence-based reagent | *Rtn4rl2* probe (mouse) RNAscope | Advanced Cell Diagnostics, | Cat# 450761 | |
| Software, algorithm | webKnossos | *Boergens et al., 2017* | https://webknossos.org | |
| Software, algorithm | Amira | Thermo Scientific, US | https://www.thermofisher.com/amira | |
| Software, algorithm | MATLAB | MathWorks | https://www.mathworks.com/products/matlab.html | |
| Software, algorithm | IGOR Pro | WaveMetrics | Version 6.3, https://www.wavemetrics.com | |
| Software, algorithm | Patchers Power Tools | http://www3.mpibpc.mpg.de/groups/neher/index.php?page=software | | |
| Software, algorithm | Imaris | Oxford Instruments (Bitplane) | Version 9.6 https://imaris.oxinst.com | |
| Software, algorithm | Fiji (ImageJ) | *Schindelin et al., 2012* https://doi.org/10.1038/nmeth.2019 | https://fiji.sc/ | |
| Software, algorithm | Python | Python Software Foundation | Version 3.8.17 https://www.python.org | |
| Software, algorithm | GraphPad Prism | GraphPad Software Inc | https://www.graphpad.com/scientific-software/prism/ | |
| Software, algorithm | Inkscape | The Inkscape Project | Version 1.3.2 https://inkscape.org/ | |

## Animals

RTN4RL2 null mutant mice on a C57BL/6N background were generated as previously described (*Wörter et al., 2009*). Knock-out (RTN4RL2 KO) and wild-type (RTN4RL2 WT) mice were derived from heterozygous matings. Both male and female mice were used in this study. Animals were genotyped as previously described (*Wörter et al., 2009*). The ages of the mice varied depending on the experiment as noted in the manuscript. The breeding and the experiments were approved by the Institutional Animal

Care and Use Committee at the Medical University of Innsbruck, local Animal Welfare Committee of the University Medical Center Göttingen and the Max Planck Institute for Multidisciplinary Sciences, as well as the Animal Welfare Office of the state of Lower Saxony, Germany (LAVES, AZ: 19/3134). The ABR experiments were carried out under the approval of the Austrian Ministry of Education, Science and Research (reference number BMWFW-66.011/0120-WF/V/3b/2016).

## Patch-clamp recordings

The apical turn of the organ of Corti was dissected from p21 to p29 animals in HEPES Hanks solution containing (in mM): 5.36 KCl, 141.7 NaCl, 10 HEPES, 0.5 MgSO$_4$, 1 MgCl$_2$, 5.6 D-glucose, and 3.4 L-glutamine (pH 7.2, ~300 mOsm/l). IHCs were exposed by gently removing nearby supporting cells by negative pressure through a glass pipette from either modiolar or pillar side. All experiments were conducted at room temperature (RT, 20–25°C). Patch pipettes were made from GB150-8P or GB150F-8P borosilicate glass capillaries (Science Products, Hofheim, Germany) for perforated and ruptured patch-clamp configurations, respectively. Pipettes were fire polished with a custom-made microforge and coated with Sylgard to minimize the capacitive noise, whenever capacitance recordings were performed. The measurements were performed using an EPC-10 amplifier controlled by Patchmaster software (HEKA Elektronik, Germany). The holding potential for IHCs was set to –87 mV across all the experiments.

## Ruptured patch-clamp

For ruptured patch-clamp, we used extracellular solution containing (in mM): 2.8 KCl, 105 NaCl, 10 HEPES, 1 CsCl, 1 MgCl$_2$, 5 CaCl$_2$, 35 TEA-Cl, and 2 mg/ml D-glucose (pH 7.2, ~300 mOsm/l). Intracellular solution contained in mM: 111 Cs-glutamate, 1 MgCl$_2$, 1 CaCl$_2$, 10 EGTA, 13 TEA-Cl, 20 HEPES, 4 Mg-ATP, 0.3 Na-GTP, and 1 L-glutathione (pH 7.3, ~290 mOsm/l). Additionally, Ca$^{2+}$ indicator Fluo4-FF (0.8 mM, Life Technologies) and TAMRA-conjugated RIBEYE/Ctbp2 binding peptide (10 mM, synthesized by the group of Prof. Dr. Olaf Jahn, Göttingen) were added to the intracellular solution for live imaging (*Zenisek et al., 2004*). Voltage dependency of Ca$^{2+}$ influx was recorded by applying 20 ms long step depolarizations with 5 mV increment. For Ca$^{2+}$ imaging, voltage ramp depolarizations ranging from –87 to 63 mV in the course of 150 ms were applied to the cells. Leak correction was performed using the p/4 protocol and liquid junction potential of 17 mV was corrected offline. Recordings were discarded from the analysis if the series resistance ($R_s$) exceeded 14 MOhm during the first 3 min after rupturing the cell, leak current exceeded –50 pA at the holding potential, and Ca$^{2+}$ current rundown was more than 25%.

Recordings were analyzed using Igor Pro 6.3 (Wavemetrics) custom-written programs. Ca$^{2+}$ current–voltage relationships (*IV* curves) were obtained by averaging approximately 5 ms segments at the maximal activation regions of individual current traces and plotting them against the depolarization voltages. Fractional activation curves of the Ca$^{2+}$ channels were calculated from the *IV*s, normalized, and fitted with the Boltzmann function: $G/G_{max} = \frac{1}{1+\exp\left[\frac{V_m - V_{half}}{k}\right]}$, where $G$ is conductance, $G_{max}$ is the maximal conductance, $V_m$ is the membrane potential, $V_{half}$ is the voltage corresponding to the half maximal activation of Ca$^{2+}$ channels, and $k$ (slope factor of the curve) is the voltage sensitivity of Ca$^{2+}$ channel activation.

## Perforated patch-clamp recordings

To perform simultaneous membrane capacitance ($C_m$) and Ca$^{2+}$ current measurements, we used extracellular solution containing (in mM): 106 NaCl, 35 TEA-Cl, 2.8 KCl, 1 MgCl$_2$, 1 CsCl, 10 HEPES, 3 CaCl$_2$, and 5.6 D-glucose (pH 7.2, ~300 mOsm/l). The pipette solution contained (in mM): 137 Cs-gluconate, 15 TEA-Cl, 3 GTP-Na$_2$, 1 ATP-Mg, 10 HEPES, 1 MgCl$_2$, as well as 300 mg/ml amphotericin B (pH 7.17, ~290 mOsm/l). Perforated patch-clamp recordings from IHCs were described previously (*Moser and Beutner, 2000*). $C_m$ changes were measured using Lindau-Neher technique (*Lindau and Neher, 1988*). To evoke exocytosis, IHCs were stimulated by step depolarizations (to –17 mV) of different durations (2, 5, 10, 20, 50, and 100 ms) in a randomized manner. Currents were leak-corrected using a p/5 protocol. Recordings were used only if the leak current was lower than 30 pA and the $R_s$ was lower than 30 MOhm. Recordings were analyzed using Igor Pro 6.3 (Wavemetrics) custom-written programs. Ca$^{2+}$ charge ($Q_{Ca^{2+}}$) was calculated by the time integral of the leak-subtracted Ca$^{2+}$ current during the depolarization step. $\triangle C_m$ was calculated as the difference between the average $C_m$ before and after

the depolarization. To measure average $C_m$, we used 400 ms segments and skipped the initial 100 ms after the depolarization.

## Ca²⁺ imaging

Ca²⁺ imaging was performed using a spinning disc confocal microscope, as described before (*Ohn et al., 2016*). Briefly, the setup was equipped with a spinning disc confocal unit (CSU22, Yokogawa) mounted on an upright microscope (Axio Examiner, Zeiss) and scientific CMOS camera (Andor Neo). Images were acquired using 63x, 1.0 NA objective (W Plan-Apochromat, Zeiss). The pixel size was measured to be 103 nm.

IHCs were loaded with Fluo4-FF Ca²⁺ dye and TAMRA-conjugated Ctbp2 binding dimeric peptide via the patch pipette. First, the cells were scanned from bottom to top by imaging TAMRA fluorescence with 561 nm laser (Jive, Cobolt AB) and exposing each plane for 0.5 s. The stack was acquired with 0.5 μm step size using Piezo positioner (Piezosystem). This allowed us to identify the planes containing synaptic ribbons. Next, we recorded Fluo4-FF fluorescence increase at individual synapses by imaging ribbon containing planes with 491 nm laser (Calypso, Cobolt AB) at 100 Hz while applying voltage ramp depolarizations to the cell. Two voltage ramps were applied at each plane, one being 5 ms shifted relative to the other. Spinning disk rotation speed was set to 2000 rpm in order to synchronize with Ca²⁺ imaging.

Images were analyzed using Igor Pro 6.3 (Wavemetrics). Ca²⁺ hotspots were identified by subtracting the average signal of several baseline frames from the average signal of 5 frames during stimulation ($\triangle F$ image). Since the same Ca²⁺ hotspot appears across multiple planes, the plane exhibiting the strongest signal was chosen. The intensities of the 3 × 3 matrix surrounding the central pixel of the hotspot were averaged across all time points to obtain the intensity profiles of Ca²⁺ influx over time. Afterward, the background signal (average of approximately 60 × 60 pixel intensities outside the cell) was subtracted from the intensity–time profiles and $\triangle F/F_0$ traces were calculated. To enhance voltage resolution of Ca²⁺ imaging, two $\triangle F/F_0$ traces corresponding to two voltage ramp depolarizations (one shifted by 5 ms over the other) were combined and plotted against the corresponding voltages (FV curves). These curves were subsequently fitted with a modified Boltzmann function. Afterward, fractional activation curves were calculated by fitting the linear decay of the fluorescence signal from the FV curves with a linear function ($G_{max}$) and then dividing the FV fit by the $G_{max}$ line. The resulting curves were further fitted with a Boltzmann function. Maximal Ca²⁺ influx ($\triangle F/F_{0\,max}$) was calculated by averaging 5 points during the stimulation.

## Immunohistochemistry

For whole-mount immunofluorescence, cochleae were fixed in 4% formaldehyde on ice for 45–60 min or sometimes overnight. For anti-Ca$_V$1.3 stainings, the cochleae were fixed for 10 min. After the fixation, the organs of Corti were microdissected in PBS and blocked in GSDB (goat serum dilution buffer; 16% goat serum, 20 mM phosphate buffer (PB), 0.3% Triton X-100, 0.45 M NaCl) for 1 hr at RT. The samples were then incubated in the primary antibody mixture overnight at 4°C. For anti-GluA4 immunolabelings, samples were incubated in the primary antibody mixture at 37°C overnight. The next day, samples were washed three times using wash buffer (20 mM PB, 0.3% Triton X-100, 0.45 M NaCl) followed by incubation in the secondary antibody mixture for 1 hr at RT. Afterwards, the samples were washed 3 times in the washing buffer, one time in PB and mounted using mounting medium (Mowiol).

The following primary antibodies were used (Key Resources Table): rabbit anti-Homer1 (1:500, 160 002, Synaptic Systems), mouse anti-Ctbp2 (1:200, 612044, BD Biosciences), rabbit anti-Ca$_V$1.3 (1:100, ACC-005, Alomone Labs), mouse anti-Bassoon (1:300, ab82958, Abcam), mouse anti-GluA2 (1:200, MAB397, Millipore), rabbit anti-GluA4 (1:200, AB1508, Millipore), rabbit anti-GluA2/3 (1:200, AB1506, Chemicon), rabbit anti-Myosin7a (1:800, ab3481, Abcam, or Proteus BioSciences), guinea pig anti-RibeyeA (1:500, 192104, Synaptic Systems), guinea pig anti-Synapsin1/2 (1:500, 106004, Synaptic Systems), guinea pig anti-Vglut3 (1:500, 135204, Synaptic Systems), chicken anti-parvalbumin (1:200, 195006, Synaptic Systems), chicken anti-calretinin (1:200, 214106, Synaptic Systems), guinea pig anti-VAChT (1:1000, 139105, Synaptic Systems), and mouse anti-ATP1A3 (1:300, MA3-915, Thermo Fisher Scientific).

The following secondary antibodies were used: Alexa Fluor 488 conjugated anti-guinea pig (1:200, A11073, Thermo Fisher Scientific), Alexa Fluor 488 conjugated anti-rabbit (1:200, A11008, Thermo

Fisher Scientific), Alexa Fluor 568 conjugated anti-chicken (1:200, ab175711, Abcam), Alexa Fluor 633 conjugated anti-guinea pig (1:200, A21105, Thermo Fisher Scientific), STAR 580 conjugated anti-mouse (1:200, Abberior, ST580P-1001-500UG), STAR 635 conjugated anti-rabbit (1:200, Abberior, ST635P-1002-500UG).

Confocal stacks were acquired using Leica SP8 confocal microscope or Abberior Instruments Expert Line STED microscope.

The volumes of the Ctbp2- and Homer1-positive puncta were estimated using the surface algorithm of Imaris software (version 9.6.0, Bitplane). The following parameters were used to create the surfaces for both Ctbp2 and Homer1 puncta in the analyzed stacks contributing to the data shown in *Figure 2*: surface detail 0.07 μm, background subtraction 0.562 μm, touching object size 0.4 μm.

The brightness and the contrast of the representative images were adjusted using Fiji (ImageJ) software for the visualization purposes.

## SGN densities

RTN4RL2 WT and RTN4RL2 KO cochlea from p40 mice of either sex were processed for SGN counts. Mouse monoclonal anti-βIII-tubulin (1:1000, 801202, Biolegend) at 4°C, followed by secondary Alexa Fluor 488 conjugated anti-mouse (1:1000, Thermo Fisher Scientific), was used to label all SGNs. Nuclei were visualized using a DAPI fluorescent counterstain (1:1000, Life Technologies). 20X images of mid-modiolar cochlear sections were collected. ImageJ was used to outline the spiral ganglion and generate the area. Quantitative assessment was performed on every other mid-modiolar section to reduce chances of double counting. Labeled cells were counted only if they had a round cell body, presence of nucleus, and homogenous cytoplasm. Densities were calculated and statistical differences were measured using a Student's *t*-test (GraphPad Prism, GraphPad Software Inc).

## Inner and outer hair cell counts

IHC and OHC counts were performed for RTN4RL2 WT and RTN4RL2 KO mice at the ages of p15, 1 month, and 2 months. Images were imported to ImageJ software, where IHC/OHC counts were performed blinded to genotype using the 'Cell counter' tool. To quantify the cells, the nuclear staining of the DAPI channel was used. The number of IHCs and OHCs was assessed along a length of 100 μm for each image, and then averaged across all samples to obtain the average number of IHCs (or OHCs) per 100 μm.

## RNAscope

RNA in situ hybridization was performed using RNAscope (Advanced Cell Diagnostics). Standard mouse probes were used to examine the expression of *Rtn4rl2* (450761, ACD) and *Gria2* (865091, ACD) closely following the manufacturer's instructions. Briefly, 6 mm paraffin tissue sections underwent deparaffinization with xylene and a series of ethanol washes. Tissues were heated in kit-provided antigen retrieval buffer and digested by kit-provided proteinase. Sections were exposed to mFISH target probes and incubated at 40°C in a hybridization oven for 2 hr. After rinsing, mFISH signal was amplified using company-provided pre-amplifier and amplifier conjugated to fluorescent dye. Subsequently, sections were blocked with 1% BSA, 2% normal goat serum in 1x PBS containing 0.3% Triton X-100 for 1 hr at RT. The tissue was incubated in mouse anti-βIII-tubulin antibody (1:500, 801202, Biolegend) overnight at 4°C. The next day, sections were rinsed with PBS, blocked again before incubating in secondary Alexa Fluor 488 conjugated anti-mouse antibody (1:1000, Invitrogen) for 2 hr at RT. Sections were counterstained with DAPI (1:1000, Life Technologies), mounted, and stored at 4°C until image analysis. mFISH images were captured on a Leica SP8 confocal microscope and processed using ImageJ.

## Serial block-face scanning electron microscopy

### Sample preparation for SBEM

One RTN4RL2 WT and two RTN4RL2 KO female mice at p36 were used for the SBEM experiments. Cochleae were processed for SBEM imaging as previously described (*Hua et al., 2021*).

In brief, the animals were decapitated after $CO_2$ inhalation under anesthesia. The cochleae were dissected from the skulls and immediately fixed by perfusing with ice-cold fixative through the round and oval windows at constant flow speed using an infusion pump (Micro4, WPI). The fixative solution

was freshly prepared and made of 4% paraformaldehyde (Sigma-Aldrich), 2.5% glutaraldehyde (Sigma-Aldrich), and 0.08 M cacodylate (pH 7.4, Sigma-Aldrich). After being immersed in the fixative at 4°C for 5 hr, the cochleae were transferred to a decalcifying solution containing the same fixative and 5% ethylenediaminetetraacetic acid (Serva) and incubated at 4°C for 5 hr. The samples were then washed twice with 0.15 M cacodylate (pH 7.4) for 30 min each, sequentially immersed in 2% $OsO_4$ (Sigma-Aldrich), 2.5% ferrocyanide (Sigma-Aldrich), and again 2% $OsO_4$ at RT for 2, 2, and 1.5 hr. After being washed in 0.15 M cacodylate and distilled water (Sartorius) for 30 min each, the samples were sequentially incubated in filtered 1% thiocarbohydrazide (Sigma-Aldrich) solution and 2% $OsO_4$ at RT for 1 and 2 hr, as well as in lead aspartate solution (0.03 M, pH 5.0, adjusted with KOH) at 50°C for 2 hr with immediate two washing steps with distilled water at RT for 30 min each. For embedding, the samples were dehydrated through graded pre-cooled acetone (Carl Roth) series (50%, 75%, and 90%, for 30 min each, all cooled at 4°C) and then pure acetone at RT (three times for 30 min each). The sample infiltration started with 1:1 and 1:2 mixtures of acetone and Spurr's resin monomer (4.1 g ERL 4221, 0.95 g DER 736, 5.9 g NSA, and 1% DMAE; Sigma-Aldrich) at RT for 6 and 12 hr on a rotator. After being impregnated in pure resin for 12 hr, the samples were placed in embedding molds (Polyscience) and hardened in a pre-warmed oven at 70°C for 72 hr.

## Sample trimming and SBEM imaging

The sample blocks were mounted upright along the conical center axis on aluminum metal rivets (3VMRS12, Gatan, UK) and trimmed coronally toward the modiolus using a diamond trimmer (TRIM2, Leica, Germany). For each sample, a block face of about ~600 × 800 mm$^2$, centered at the apical segment based on the anatomical landmarks, was created using an ultramicrotome (UC7, Leica, Germany). The samples were coated with a 30-nm thick gold layer using a sputter coater (ACE600, Leica, Germany). The serial images were acquired using a field-emission scanning EM (Gemini300, Carl Zeiss, Germany) equipped with an in-chamber ultramicrotome (3ViewXP, Gatan, UK) and back-scattered electron detector (Onpoint, Gatan, UK). Focal charge compensation was set to 100% with a high vacuum chamber pressure of $2.8 \times 10^3$ mbar. The following parameters were set for the SBEM imaging: 12 nm pixel size, 50 nm nominal cutting thickness, 2 keV incident beam energy, and 1.5 ms pixel dwell time.

For the RTN4RL2 WT dataset, 2377 consecutive slices (9000 × 15,000 pixels) were collected, whereas the two RTN4RL2 KO datasets had 3217 slices (16,000 × 10,000 pixels) and 2425 slices (9000 × 15,000 pixels). All datasets were aligned along the z-direction using a custom MATLAB script based on cross-correlation maximum between consecutive slices (*Hua et al., 2022*) before being uploaded to webKnossos (*Boergens et al., 2017*) for skeleton and volume tracing.

## Identification and quantification of auditory afferent fibers

In our SBEM datasets of the cochlea, manual skeleton tracing was carried out on all neurites that originated from three neighboring HP at the center of each dataset. To search for type I afferent fibers, several morphological features were used, such as myelination after entering HP, radial and unbranched fiber trajectory, contact with IHCs, as well as ribbon-associated terminals (*Hua et al., 2021*). This resulted in 115 putative type I afferent fibers, and further classification was made based on the presence of presynaptic ribbon, fiber branching, and contact with IHC. For illustration purposes, a type I afferent fiber bundle of one RTN4RL2 KO dataset was volume traced and 3D rendered using Amira software (Thermo Scientific, US).

## Ribbon size measurement and synapse counting

Ribbon-type synapses were manually annotated in 18 intact IHCs captured by SBEM using webKnossos. The dense core region of individual ribbon synapses was manually contoured, and the associated voxels were counted for ribbon volume measurement. In the case of multi-ribbon synapses, all ribbon bodies at a single AZ were summed up to yield the ribbon volume.

## **Auditory brainstem responses**

ABRs were recorded in 2- to 4-month-old mice of either sex, as described previously (*Luque et al., 2021*). Briefly, we used a custom-made system in an anechoic chamber in a calibrated open-field configuration. ABRs were recorded via needle electrodes in response to tone bursts of 4, 8, 16, and

32 kHz or a click wide spectrum (2–45.2 KHz, 2 KHz steps) as stimuli. Tone pips of 3 ms duration (1 ms rise and fall time) were presented at a rate of 60/s with alternating phases. Starting with 0 dB, the stimuli were increased in 5 dB steps up to 120 dB. Hearing thresholds were determined as the minimum stimulation level that produced a clearly recognizable potential. Both ears were tested. Since there was no significant difference between the right and left ears, we did not consider this factor further. Recordings were evaluated by three independent researchers in a blinded manner.

## Data analysis and statistics

The data were analyzed using Pro (Wavemetrics), Python, and GraphPad Prism (GraphPad Software Inc) software. For two sample comparisons, data were tested for normality and equality of variances using Jarque–Bera and $F$-test, respectively. Afterward, two-tailed Student's $t$-test or Mann–Whitney–Wilcoxon test was performed. The latter was used when normality and/or equality of variances were not met. For ABR thresholds Kruskal–Wallis test followed by Dunn's multiple comparison test was used. Data is presented as mean ± SEM, unless otherwise stated. Mean, SEM, and SD are indicated in the figure captions as mean ± SEM, SD. Significances are reported as $*p < 0.05$, $**p < 0.01$, $***p < 0.001$. The number of the animals is indicated as $N$.

## Acknowledgements

We thank Sina Langer, Christiane Senger-Freitag, and Sandra Gerke for expert technical support. We thank Prof. Dr. Carolin Wichmann, Dr. Susann Michanski, Julius Bahr, and Sophia Mutschall for the help with the SBEM sample preparation. We thank Dr. Yi Jiang and Haoyu Wang for the visualization of neurite reconstruction. We further thank Dr. Mark Rutherford for providing tips for GluA4 immunolabeling. We would also like to thank Prof. Dr. Olaf Jahn and Lars van Werven for the TAMRA-conjugated Ctbp2-binding peptide synthesis. The study was supported by the German Research Foundation through the Cluster of Excellence (EXC2067) Multiscale Bioimaging (TM) and the Leibniz Program (MO896/5 to TM) as well as by the European Research Council through the Advanced Grant 'DynaHear' to TM under the European Union's Horizon 2020 Research and Innovation program (grant agreement No. 101054467), and by Fondation Pour l'Audition (FPA RD-2020-10), by the SPIN-FWF grant to CB, and by the National Natural Science Foundation of China (82171133 to YH), Industrial Support Fund of Huangpu District in Shanghai (XK2019011 to YH), Innovative Research Team of High-level Local Universities in Shanghai (SHSMU-ZLCX20211700). NK is a member of the Hertha Sponer College from the Cluster of Excellence Multiscale Bioimaging (MBExC). TM is a Max-Planck Fellow at the Max Planck Institute for Multidisciplinary Sciences. Open access funding provided by Max Planck Society.

## Additional information

### Funding

| Funder | Grant reference number | Author |
| --- | --- | --- |
| Deutsche Forschungsgemeinschaft | Germany's Excellence Strategy - EXC2067 | Tobias Moser |
| Deutsche Forschungsgemeinschaft | Leibniz Program (MO896/5) | Tobias Moser |
| European Research Council | "DynaHear" (grant agreement No. 101054467) | Tobias Moser |
| Fondation Pour l'Audition | FPA RD-2020-10 | Tobias Moser |
| FWF Austrian Science Fund | SPIN-FWF | Christine Bandtlow |
| National Natural Science Foundation of China | 82171133 | Yunfeng Hua |

| Funder | Grant reference number | Author |
|---|---|---|
| Industrial Support Fund of Huangpu District in Shanghai | XK2019011 | Yunfeng Hua |
| Innovative Research Team of High-level Local Universities in Shanghai | SHSMU-ZLCX20211700 | Yunfeng Hua |

The funders had no role in study design, data collection, and interpretation, or the decision to submit the work for publication. Open access funding provided by Max Planck Society.

## Author contributions

Nare Karagulyan, Conceptualization, Data curation, Formal analysis, Validation, Investigation, Visualization, Methodology, Writing – original draft, Writing – review and editing; Maja Überegger, Data curation, Formal analysis, Validation, Investigation, Visualization, Methodology; Yumeng Qi, Data curation, Formal analysis, Validation, Investigation, Visualization, Methodology, Writing – review and editing; Norbert Babai, Florian Hofer, Data curation, Formal analysis, Investigation, Visualization, Methodology, Writing – review and editing; Lejo Johnson Chacko, Maria Luque, Formal analysis, Validation, Methodology; Fangfang Wang, Data curation, Formal analysis, Investigation, Methodology; Rudolf Glueckert, Resources, Data curation, Formal analysis, Supervision, Validation, Investigation, Methodology, Writing – review and editing; Anneliese Schrott-Fischer, Resources, Supervision; Yunfeng Hua, Conceptualization, Resources, Data curation, Formal analysis, Supervision, Funding acquisition, Validation, Investigation, Methodology, Writing – original draft, Project administration, Writing – review and editing; Tobias Moser, Conceptualization, Resources, Supervision, Funding acquisition, Validation, Writing – original draft, Project administration, Writing – review and editing; Christine Bandtlow, Conceptualization, Resources, Supervision, Funding acquisition, Validation, Project administration, Writing – review and editing

## Author ORCIDs

Nare Karagulyan ![ORCID] https://orcid.org/0009-0005-3999-2427
Yumeng Qi ![ORCID] https://orcid.org/0000-0003-0196-6846
Tobias Moser ![ORCID] https://orcid.org/0000-0001-7145-0533
Christine Bandtlow ![ORCID] https://orcid.org/0000-0001-7437-8864

## Ethics

The breeding and the experiments were approved by the Institutional Animal Care and Use Committee at the Medical University of Innsbruck, local Animal Welfare Committee of the University Medical Center Göttingen and the Max Planck Institute for Multidisciplinary Sciences, as well as the Animal Welfare Office of the state of Lower Saxony, Germany (LAVES, AZ: 19/3134). The ABR experiments were carried out under the approval of the Austrian Ministry of Education, Science and Research (reference number BMWFW-66.011/0120-WF/V/3b/2016).

Reviewer #1 (Public review): https://doi.org/10.7554/eLife.103481.3.sa1
Reviewer #3 (Public review): https://doi.org/10.7554/eLife.103481.3.sa2
Author response https://doi.org/10.7554/eLife.103481.3.sa3

# Additional files

## Supplementary files

MDAR checklist

## Data availability

Source data files are provided with the figures. Routines for the analysis of $Ca^{2+}$ imaging data and whole-cell capacitance measurements are available as Source codes 2 and 3 of *Jean et al., 2018*.

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
