## [Editor Report · eLife Assessment]

In this work, the authors characterize the synaptic adhesion molecule RTN4RL2, demonstrating its critical involvement in the development and function of auditory synapses between inner hair cells and spiral ganglion neurons. This study is **important** because it offers potential insights into therapeutic strategies for hearing loss associated with synaptic dysfunction. The findings are **solid**, because they are supported by the use of multiple advanced techniques, including FISH and SBEM imaging.

---

## [Referee Report · Reviewer #1 (Public review)]

Hearing and balance rely on specialized ribbon synapses that transmit sensory stimuli between hair cells and afferent neurons. Synaptic adhesion molecules that form and regulate transsynaptic interactions between inner hair cells (IHCs) and spiral ganglion neurons (SGNs) are crucial for maintaining auditory synaptic integrity and, consequently, for auditory signaling. Synaptic adhesion molecules such as neurexin-3 and neuroligin-1 and -3 have recently been shown to play vital roles in establishing and maintaining these synaptic connections (doi: 10.1242/dev.202723 and DOI: 10.1016/j.isci.2022.104803). However, the full set of molecules required for synapse assembly remains unclear.

Karagulan et al. highlight the critical role of the synaptic adhesion molecule RTN4RL2 in the development and function of auditory afferent synapses between IHCs and SGNs, particularly regarding how RTN4RL2 may influence synaptic integrity and receptor localization. Their study shows that deletion of RTN4RL2 in mice leads to enlarged presynaptic ribbons and smaller postsynaptic densities (PSDs) in SGNs, indicating that RTN4RL2 is vital for synaptic structure. Additionally, the presence of "orphan" PSDs-those not directly associated with IHCs-in RTN4RL2 knockout mice suggests a developmental defect in which some SGN neurites fail to form appropriate synaptic contacts, highlighting potential issues in synaptic pruning or guidance. The study also observed a depolarized shift in the activation of CaV1.3 calcium channels in IHCs, indicating altered presynaptic functionality that may lead to impaired neurotransmitter release. Furthermore, postsynaptic SGNs exhibited a deficiency in GluA2/3 AMPA receptor subunits, despite normal Gria2 mRNA levels, pointing to a disruption in receptor localization that could compromise synaptic transmission. Auditory brainstem responses showed increased sound thresholds in RTN4RL2 knockout mice, indicating impaired hearing related to these synaptic dysfunctions.

The findings reported here significantly enhance our understanding of synaptic organization in the auditory system, particularly concerning the molecular mechanisms underlying IHC-SGN connectivity. The implications are far-reaching, as they not only inform auditory neuroscience but also provide insights into potential therapeutic targets for hearing loss related to synaptic dysfunction.

Comments on the latest version:

In the revised manuscript, the authors have addressed my previous comments and incorporated my recommendations by adding missing experimental details, using color-blind-friendly figure colors, and discussing the differences between GluA3 KO and RTN4RL2 KO phenotypes. They also clarified why the animals needed for additional experiments are no longer available. Although these specific animals are unavailable, the authors made an effort to address my concerns by performing

---

## [Referee Report · Reviewer #3 (Public review)]

In this study, the authors used RNAscope to explore the expression of RTN4RL2 RNA in hair cells and spiral ganglia. Through RTN4RL2 gene knockout mice, they demonstrated that the absence of RTN4RL2 leads to pre-synaptic changes of an increase in the size of presynaptic ribbons and a depolarized shift in the activation of calcium channels in inner hair cells. Additionally, they observed a post-synaptic reduction in GluA2-4 AMPA receptors and identified additional "orphan PSDs" not paired with presynaptic ribbons via immunostaining and an increased number of type I SGNs that are not connected with a ribbon synapse via serial block face imaging. These synaptic alterations ultimately resulted in an increased hearing threshold in mice, confirming that the RTN4RL2 gene is essential for normal hearing. These data are intriguing as they suggest that RTN4RL2 contributes to the proper formation and function of auditory afferent synapses and is critical for normal hearing. Most strikingly, the post-synaptic changes and hearing threshold changes are similar to recently published results by Carlton et al, 2024 on a mutation in Bai1, which is a potential binding partner for RTN4RL2. Overall this work provides some clues to the function of RTN4RL2 in the cochlea, but further studies are required to elucidate the function.

A few points would improve the manuscript and the strength of the data presented.

(1) A quantitative assessment is necessary in Figure 1 when discussing RNA scope data. It would be beneficial to show that expression levels are quantitatively reduced in KO mice compared to wild-type mice. This suggestion also applies to Figure 3D, which examines expression levels of Gria2. Data is provided for KO reduction in SGN, but not showing that hair cell labeling is specific. If slides are not available for the young ages, showing hair cell expression at P40 would be sufficient along with a loss of labeling at in the KO at P40.

(2) In Figure 2, the authors present a morphological analysis of synapses and discuss the presence of "orphan PSDs." I agree that Homer1 not juxtaposed with Ctbp2 is increased in KO mice compared to the control group. However, in quantifying this, they opted to measure the number of Ctbp2 puncta with Homer 1 juxtaposed, which indicates the percentages of orphan ribbons rather than directly quantifying the number of Homer1 not juxtaposed with Ctbp2. Quantifying the number of Homer1 not juxtaposed with Ctbp2 would more clearly represent "orphan PSDs" and provide stronger support for the discussion surrounding their presence. A measurement of these was provided in the rebuttal letter, and while this number much more clearly demonstrates the increase in the number of orphan puncta, this analysis is not provided in the manuscript. This number also suggests the number of orphan receptors may be quite high, outnumbering ribbons 2:1.

(3) In Figure 3, the authors discuss GluA2/3 puncta reduction and note that Gria2 RNA expression remains unchanged. However, the GluA2/3 labeling is done at 1-1.5 months, whereas the Gria2 RNAscope is done at P4. Additionally, there is a lack of quantification for Gria2 RNA expression due to their tissue being processed separately. RNA scope at a comparable age to the GluA2/3 would be stronger support for their statement that Gria2 expression is comparable despite a reduction in GluA2/3 puncta.

(4) In Figure 4, the authors indicate that RTN4RL2 deficiency reduces the number of type 1 SGNs connected to ribbons. Given that the number of ribbons remains unchanged (Figure 2), it is important to clearly explain the implications of this finding. It is already known that each type I SGN forms a single synaptic contact with a single IHC. The fact that the number of ribbons remains constant while additional "orphan PSDs" are present suggests that the overall number of SGNs might need to increase to account for these findings, however, the authors noted no change in the number of SGN soma. This discrepancy is important to point out.

---

## [Author Response]

The following is the authors’ response to the original reviews

**PublicReviews:**

**Reviewer#1 (Public review):**
Hearing and balance rely on specialized ribbon synapses that transmit sensory stimuli between hair cells and afferent neurons. Synaptic adhesion molecules that form and regulate transsynaptic interactions between inner hair cells (IHCs) and spiral ganglion neurons (SGNs) are crucial for maintaining auditory synaptic integrity and, consequently, for auditory signaling. Synaptic adhesion molecules such as neurexin-3 and neuroligin-1 and -3 have recently been shown to play vital roles in establishing and maintaining these synaptic connections (doi: 10.1242/dev.202723 and DOI: 10.1016/j.isci.2022.104803). However, the full set of molecules required for synapse assembly remains unclear.Karagulan et al. highlight the critical role of the synaptic adhesion molecule RTN4RL2 in the development and function of auditory afferent synapses between IHCs and SGNs, particularly regarding how RTN4RL2 may influence synaptic integrity and receptor localization. Their study shows that deletion of RTN4RL2 in mice leads to enlarged presynaptic ribbons and smaller postsynaptic densities (PSDs) in SGNs, indicating that RTN4RL2 is vital for synaptic structure. Additionally, the presence of "orphan" PSDs-those not directly associated with IHCs-in RTN4RL2 knockout mice suggests a developmental defect in which some SGN neurites fail to form appropriate synaptic contacts, highlighting potential issues in synaptic pruning or guidance. The study also observed a depolarized shift in the activation of CaV1.3 calcium channels in IHCs, indicating altered presynaptic functionality that may lead to impaired neurotransmitter release. Furthermore, postsynaptic SGNs exhibited a deficiency in GluA2/3 AMPA receptor subunits, despite normal Gria2 mRNA levels, pointing to a disruption in receptor localization that could compromise synaptic transmission. Auditory brainstem responses showed increased sound thresholds in RTN4RL2 knockout mice, indicating impaired hearing related to these synaptic dysfunctions.The findings reported here significantly enhance our understanding of synaptic organization in the auditory system, particularly concerning the molecular mechanisms underlying IHC-SGN connectivity. The implications are far-reaching, as they not only inform auditory neuroscience but also provide insights into potential therapeutic targets for hearing loss related to synaptic dysfunction.

We would like to thank the reviewer for appreciating the work and the advice that helped us to further improve the manuscript. We have carefully addressed all concerns, please see our point-per-point response below and the revised manuscript.

**Reviewer#2 (Public review):**
Summary:Kargulyan et al. investigate the function of the transsynaptic adhesion molecule RTN4RL2 in the formation and function of ribbon synapses between type I spiral ganglion neurons (SGNs) and inner hair cells. For this purpose, they study constitutive RTN4RL2 knock-out mice. Using immunohistochemistry, they reveal defects in the recruitment of protein to ribbon synapses in the knockouts. Serial block phase EM reveals defects in SGN projections in mutants. Electrophysiological recordings suggest a small but statistically significant depolarized shift in the activation of Cav1.3 Ca^2+^ channels. Auditory thresholds are also elevated in the mutant mice. The authors conclude that RTN4RL2 contributes to the formation and function of auditory afferent synapses to regulate auditory function.

We would like to thank the reviewer for appreciating the work and the advice that helped us to further improve the manuscript. We have carefully addressed all concerns, please see our point-per-point response below and the revised manuscript.

Strengths:The authors have excellent tools to analyze ribbon synapses.Weaknesses:However, there are several concerns that substantially reduce my enthusiasm for the study.(1) The analysis of the expression pattern of RTN4RL2 in Figure 1 is incomplete. The authors should show a developmental time course of expression up into maturity to correlate gene expression with major developmental milestones such as axon outgrowth, innervation, and refinement. This would allow the development of models supporting roles in axon outgrowth versus innervation or both.

We agree that it would be valuable to show the developmental time course of RTN4RL2 expression. In response to the reviewer’s comment, we are providing RNAscope data from developmental ages E11.5, E12.5 and E16 in Figure 1. RTN4RL2 shows expression at E11.5/E12.5 both in the spiral ganglion and hair cell region, with first onset in the hair cells. We conclude that RTN4RL2 is expressed highest during fiber growth at embryonic stages and is downregulated during postnatal development maintaining low levels of expression during adulthood.

(2) It would be important to improve the RNAscope data. Controls should be provided for Figure 1B to show that no signal is observed in hair cells from knockouts. The authors apparently already have the sections because they analyzed gene expression in SGNs of the knock-outs (Figure 1C).

In Figure 1C gene expression in SGNs was assessed at p40, while the expression in hair cells is provided for p1 animals. Unfortunately, we do not have KO controls for p1 animals. However, as indicated in our manuscript, previously published RNA expression datasets do find RTN4RL2 expression in hair cells. Therefore, we think it is unlikely that our results are unspecific.

(3) It is unclear from the immunolocalization data in Figure 1D if all type I SGNs express RTN4RL2. Quantification would be important to properly document the presence of RTN4RL2 in all or a subset of type I SGNs. If only a subset of SGNs express RTN4RL2, it could significantly affect the interpretation of the data. For example, SGNs selectively projecting to the pillar or modiolar side of hair cells could be affected. These synapses significantly differ in their properties.

According to already published single cell RNAseq dataset from Shrestha et al., 2018, RTN4RL2 expression does not seem to show a clear type I SGN subtype specificity (Author response image 1). In response to the reviewer’s comment, we have further performed anti-Parvalbumin (PV) and anti-calretinin (CR) immunostainings in mid-modiolar cryosections of RTN4RL2^+/+^ and RTN4RL2^-/-^ cochleae. Parvalbumin was chosen to label all SGNs and CALB2 was chosen primarily as a type Ia SGN marker (Sun et al., 2018). We present the data from all analyzed samples below (figure 2 of this rebuttal letter). Cell segmentation masks of PV positive cells were obtained using Cellpose 2.0 and the average CR intensity was calculated in those masks. While the distributions of CR intensity and the ratio of CR and PV intensities are slightly shifted in RTN4RL2^-/-^ cochleae, we take the data to suggest that the composition of the spiral ganglion by molecular type I SGN subtypes is largely unchanged in RTN4RL2^-/-^ mice.

Author response image 1 cites single cell RNAseq data of Brikha R Shrestha, Chester Chia, Lorna Wu, Sharon G Kujawa, M Charles Liberman, Lisa V Goodrich. Sensory neuron diversity in the inner ear is shaped by activity. Cell. 2018 Aug 23; 174(5):1229-1246.e17. doi: 10.1016/j.cell/2018.07.007

**Author response image 2. sa3fig2:** Calretinin intensity distribution in spiral ganglion of RTN4RL2^+/+^ and RTN4RL2^-/-^ mice. (A) Mid-modiolar cochlear cryosections from RTN4RL2^+/+^ (top) and RTN4RL2^-/-^ (bottom) mice immunolabeled against Parvalbumin (PV) and Calretinin (CR). Scale bar = 20 mm. (B) Distribution of CR intensity in PV positive cells (N = 3 for each genotype). (C) Distribution of the ratio of CR and PV intensities (N = 3 for each genotype).

(4) It is important to show proper controls for the RTN4RL2 immunolocalization data to show that no staining is observed in knockouts.

Unfortunately, our recent attempts to perform RTN4RL2 immunostainings on cryosections failed and therefore, we decided to remove the RTNr4RL2 immunostainings from Figure 1. We have adjusted the results section accordingly.

(5) The authors state in the discussion that no staining for RTN4RL2 was observed at synaptic sites. This is surprising. Did the authors stain multiple ages? Was there perhaps transient expression during development? Or in axons indicative of a role in outgrowth, not synapse formation?

We thank the reviewer for the comment. We have now tried RTN4RL2 immunostainings on cryosections at several developmental stages, but unfortunately this time did not succeed to obtain reproducible and reliable results. Therefore, we decided to also remove the previous immunostainings from Figure 1. We have adjusted the results section as well as removed our statement of not detecting RTN4RL2 near the synaptic regions from the discussion.

(6) In Figure 2 it seems that images in mutants are brighter compared to wildtypes. Are exposure times equivalent? Is this a consistent result?

Yes, the samples were prepared in parallel, imaged and analyzed in the same manner.

No, we did not observe consistent differences in brightness and also did not find it in the exemplary images of figure 2.

(7) The number of synaptic ribbons for wildtype in Figure 2 is at 10/IHCs, and in Figure 2 Supplementary Figure 2 at 20/IHCs (20 is more like what is normally reported in the literature). The value for mutant similarly drastically varies between the two figures. This is a significant concern, especially because most differences that are reported in synaptic parameters between wild-type and mutants are far below a 2-fold difference.

The key message is that there is no difference in the numbers of ribbons and synapses between the genotypes for the cochlear apex (~10 ribbons/IHCs, Figure 2 and Figure 2-figure supplement 2) and the mid- and base of the cochlea (more ribbons/IHCs, Figure 2-figure supplement 2). Figure 2-figure supplement 3 (now Figure 3) shows that there is a massive reduction of postsynaptic GluA2, while both Figure 2 and Figure 2-figure supplement 2 indicate that the number synapses is normal. These are two different data sets and while we closely collaborated and also shared the Moser lab protocols and analysis routines, we agree that there is a difference in the absolute synapse count, which most likely was an observer difference and different choice of tonotopic positions of analysis. In Figure 2 only the apical hair cells have been analyzed. The Moser lab, since establishing the immunofluorescence-based quantification of synapse number (Khimich et al., 2005) reported tonotopic differences in synapse counts (focus of Meyer et al., 2009 and reported by others: e.g. Kujawa and Liberman, 2009): apical and basal IHCs lower synapse numbers than mid-cochlear IHCs.

(8) The authors report differences in ribbon volume between wild-type and mutant. Was there a difference between the modiolar/pillar region of hair cells? It is known that synaptic size varies across the modiolar-pillar axis. Maybe smaller synapses are preferentially lost?

We thank the reviewer for the comment. Unfortunately, our already acquired datasets from 3-week-old mice did not allow us to check whether the previously described modiolar-pillar gradient of the ribbon size was collapsed in RTN4RL2^-/-^ mice due to the not so well-preserved morphology of the inner hair cells in our preparations. However, since the number of the ribbons is not changed in the RTN4RL2 KO mice, we do not think that the increase in the ribbon size is due to the loss of small ribbons. In response to the reviewers comment we have analyzed the modiolar-pillar gradient of the ribbon size in IHCs of middle turn of the cochlea form a newly acquired dataset of 14-week-old mice. We took the fluorescence intensity of Ctbp2 positive puncta as a proxy for the ribbon size. In these older mice we found a preserved modiolar-pillar gradient of the ribbon size (larger ribbons at the modiolar side). We summarized the results in the below Author response image 3.

**Author response image 3. sa3fig3:** The modiolar-pillar gradient of ribbon size is preserved in RTN4RL2^-/-^ IHCs. (**A**) Maximum intensity projections of approximately 2 IHCs stained against Vglut3 and Ctbp2 from 14-week-old RTN4RL2^+/+^ (left) and RTN4RL2^-/-^ (right) mice. Scale bar = 5 mm. (**B**) Synaptic ribbons on the modiolar side show higher fluorescence intensity than the ones on the pillar side of mid-cochlear IHCs in both RTN4RL2^+/+^ (left, N=2) RTN4RL2^-/-^ (right, N=2) mice. (**C**) Average fluorescence intensity of modiolar ribbons per IHC is higher than the average fluorescence intensity of pillar ribbons (paired t-test, p < 0.001).

(9) The authors show in Figure 2 - Supplement 3 that GluA2/3 staining is absent in the mutants. Are GluA4 receptors upregulated? Otherwise, synaptic transmission should be abolished, which would be a dramatic phenotype. Antibodies are available to analyze GluA4 expression, the experiment is thus feasible. Did the authors carry out recordings from SGNs?

In response to the reviewer’s comment, we have performed GluA4 stainings in RTN4LR2^-/-^ mice and did not detect any GluA4 positive signal in the mutants (new Figure 3-figure supplement 1). Unfortunately, our animal breeding license was expired at the time we received the reviews and that is why our results are from 14-week-old animals. To verify that the absence of GluA4 signal is not due to potential PSD loss in 14-week-old RTN4RL2^-/-^, we have additionally performed anti-Ctbp2, anti-Homer1 and anti-Vglut3 stainings in 14-week-old animals. Despite the reduced number, we still observed juxtaposing pre- and postsynaptic puncta. We assume that the reviewer asks for patch-clamp recordings from SGNs, which are, as we are confident the reviewer is aware of, technically very challenging and beyond the scope of the present study but an important objective for future studies. In response to the reviewers comment we have added a statement to the discussion pointing to these patch-clamp recordings from SGNs as important objective for future studies.

(10) The authors use SBEM to analyze SGN projections and synapses. The data suggest that a significant number of SGNs are not connected to IHCs. A reconstruction in Figure 3 shows hair cells and axons. It is not clear how the outline of hair cells was derived, but this should be indicated. Also, is this a defect in the formation of synapses and subsequent retraction of SGN projections? Or could RTN4RL2 mutants have a defect in axonal outgrowth and guidance that secondarily affects synapses? To address this question, it would be useful to sparsely label SGNs in mutants, for example with AAV vectors expression GFP, and to trace the axons during development. This would allow us to distinguish between models of RTN4RL2 function. As it stands, it is not clear that RTN4RL2 acts directly at synapses.

We agree with the reviewer on the value of a developmental study of afferent connectivity but consider this beyond the scope of the present study. In response to the reviewer's comment, we have replaced the IHC outlines with volume-reconstructed IHCs in Figure 3B (now Figure 4B). Moreover, as shown in Figure 3F (now Figure 4F), most if not all type-I SGNs (both with and without ribbon) were unbranched in the mutants just like in wildtype (also shown for a larger sample in Hua et al., 2021), arguing against morphological abnormality during development.

(11) The authors observe a tiny shift in the operation range of Ca^2+^ channels that has no effect on synaptic vesicle exocytosis. It seems very unlikely that this difference can explain the auditory phenotype of the mutant mice.

We assume that the statement refers to the normal exocytosis of mutant IHCs at the potential of maximal Ca^2+^ influx (Figure 3G and H, now Figure 4G and H). We would like to note that this experiment was performed to probe for a deficit of synapse function beyond that of the Ca^2+^ channel activation, but did not address the impact of the altered voltage—dependence of Ca^2+^ channel activation. In response to the reviewer’s comment, we have now added further discussion to more clearly communicate that for the range of receptor potentials achieved near sound threshold we expect impaired IHC exocytosis as the Ca^2+^ channels require slightly more depolarization for activation in the mutant IHCs.

(12) ABR recordings were conducted in whole-body knockouts. Effects on auditory thresholds could be a secondary consequence of perturbation along the auditory pathway. Conditional knockouts or precisely designed rescue experiments would go a long way to support the authors' hypothesis. I realize that this is a big ask and floxed mice might not be available to conduct the study.

Thanks for this helpful comment and, indeed, unfortunately, we do not have conditional KO mice at our disposal. We totally agree that this will be important also for clarifying the role of IHC vs. SGN expression of RTN4RL2. In response to the reviewer’s comment, we now discussed the shortcoming of using constitutive RTN4RL2^-/-^ mice and added this important experiment on IHC and SGN specific deletion of RTN4RL2 as an objective of future studies.

**Reviewer #3 (Public review):**
In this study, the authors used RNAscope and immunostaining to confirm the expression of RTN4RL2 RNA and protein in hair cells and spiral ganglia. Through RTN4RL2 gene knockout mice, they demonstrated that the absence of RTN4RL2 leads to an increase in the size of presynaptic ribbons and a depolarized shift in the activation of calcium channels in inner hair cells. Additionally, they observed a reduction in GluA2/3 AMPA receptors in postsynaptic neurons and identified additional "orphan PSDs" not paired with presynaptic ribbons. These synaptic alterations ultimately resulted in an increased hearing threshold in mice, confirming that the RTN4RL2 gene is essential for normal hearing. These data are intriguing as they suggest that RTN4RL2 contributes to the proper formation and function of auditory afferent synapses and is critical for normal hearing. However, a thorough understanding of the known or postulated roles of RTN4Rl2 is lacking.

We would like to thank the reviewer for appreciating the work and the advice that helped us to further improve the manuscript. We have carefully addressed all concerns, please see our point-per-point response below and the revised manuscript.

While the conclusions of this paper are generally well supported by the data, several aspects of the data analysis warrant further clarification and expansion.(1) A quantitative assessment is necessary in Figure 1 when discussing RNA and protein expression. It would be beneficial to show that expression levels are quantitatively reduced in KO mice compared to wild-type mice. This suggestion also applies to Figure 2-supplement 3.D, which examines expression levels.

The processing of our control and KO samples for RNAscope was not strictly done in parallel and therefore we would like to refrain from quantitative comparison.

(2) In Figure 2, the authors present a morphological analysis of synapses and discuss the presence of "orphan PSDs." I agree that Homer1 not juxtaposed with Ctbp2 is increased in KO mice compared to the control group. However, in quantifying this, they opted to measure the number of Homer1 juxtaposed with Ctbp2 rather than directly quantifying the number of Homer1 not juxtaposed with Ctbp2. Quantifying the number of Homer1 not juxtaposed with Ctbp2 would more clearly represent "orphan PSDs" and provide stronger support for the discussion surrounding their presence.

We appreciate the reviewer’s comment. We did not perform this analysis primarily because “orphan” Homer1 puncta, as seen in our immunostainings, are distributed away from hair cells in diverse morphologies and sizes. This makes distinguishing them from unspecific immunofluorescent spots—also present in wild-type samples—challenging. In response to the reviewer’s request, we analyzed the number of “orphan” Homer1 puncta in our previously acquired RTN4RL2^+/+^ and RTN4RL2^-/-^ samples. Using the surface algorithm in Imaris software, we applied identical parameters across all samples to create surfaces for Homer1-positive puncta (total Homer1 puncta). We quantified “orphan” Homer1 puncta as the difference between total and ribbon-juxtaposing Homer1 puncta and normalized this number to the IHC count. Our results showed 4.3 vs. 26.8 “orphan” Homer1 puncta per IHC in RTN4RL2^+/+^ and RTN4RL2^-/-^ samples, respectively. We note that variations in acquired volumes between samples may introduce confounding effects.

(3) In Figure 2, Supplementary 3, the authors discuss GluA2/3 puncta reduction and note that Gria2 RNA expression remains unchanged. However, there is an issue with the lack of quantification for Gria2 RNA expression. Additionally, it is noted that RNA expression was measured at P4. While the timing for GluA2/3 puncta assessment is not specified, if it was assessed at 3 weeks old as in Figure 2's synaptic puncta analysis, it would be inappropriate to link Gria2 RNA expression with GluA2/3 protein expression at P4. If RNA and protein expression were assessed at P4, please indicate this timing for clarity.

GluA2/3 immunostainings were performed in 1 to 1.5-month-old animals. We apologize for not indicating this before and have now included it in Figure 3 legend. The processing of our control and KO samples for RNAscope was not strictly done in parallel and therefore we would like to refrain from quantitative comparison.

(4) In Figure 3, the authors indicate that RTN4RL2 deficiency reduces the number of type 1 SGNs connected to ribbons. Given that the number of ribbons remains unchanged (Figure 2), it is important to clearly explain the implications of this finding. It is already known that each type I SGN forms a single synaptic contact with a single IHC. The fact that the number of ribbons remains constant while additional "orphan PSDs" are present suggests that the overall number of SGNs might need to increase to account for these findings. An explanation addressing this would be helpful.

In Figure 3 (now Figure 4), we found additional type-1 SGNs that are unconnected to IHC, in good agreement with “orphan PSDs” observed under the light microscope. Indeed, we also confirmed monosynaptic, unbranched fiber morphology (Figure 3F, now Figure 4F). Together, these results imply about a 20% increase in the overall number of SGNs, which however we did not observe in SGN soma counting.

(5) In Figure 4F and 5Cii, could you clarify how voltage sensitivity (k) was calculated? Additionally, please provide an explanation for the values presented in millivolts (mV).

Voltage sensitivity (k) was calculated as the slope of the Boltzmann fit to the fractional activation curves: \begin{document}$G / G_{\max }=\frac{1}{1+\exp \left[\frac{V_{m}-V_{\text {half }}}{k}\right]}$\end{document}, Where *G* is conductance, *Gmax* is the maximum conductance, *Vm* is the membrane potential, *Vhalf* is the voltage corresponding to the half maximal activation of Ca^2+^ channels and *k* (slope of the curve) is the voltage sensitivity of Ca^2+^ channel activation. We have now added this to our Materials and Methods section.

(6) In Figure 6, the author measured the threshold of ABR at 2-4 months old. Since previous figures confirming synaptic morphology and function were all conducted on 3-week-old mice, it would be better to measure ABR at 3 weeks of age if possible.

ABR measurements for comparisons in a cohort of age-matched mice require fully developed individuals. 3 weeks is the minimum age that is regarded for a mature ear. However, variation in developmental differences among one litter is very frequent that affects normal hearing thresholds. From our own experience we do not regard the ear fully functional before 6 weeks of age. Then hearing thresholds are lowest indicating full functionality. Since the C57BL/6 background strain has a genetic defect in the Cadherin 23-coding gene (Cdh23) at the ahl locus of mouse chromosome 10 these mice exhibit early onset and progression of age-related hearing loss starting at 5–8 months (Hunter & Willott, 1987). Therefore, we chose a “safe” time window for stable and unaffected ABR recordings of 2-4 months to provide most representative data.

**Recommendations for the authors:**

**Reviewer #1 (Recommendations for the authors):**
(1) Please include information on the validation of all the antibodies used in this study, or reference the relevant work where the antibodies were previously validated.

In response to the reviewer’s comment, we have now included a table listing all primary antibodies used in this study. Where possible, we provide references for knockout (KO) validation. Otherwise, we refer to the manufacturer’s information, as provided in the respective datasheets.

(2) Figure 2 illustrates the pre- and postsynaptic changes observed in RTN4RL2 knockout (KO) mice. Please specify the age of the mice and the cochlear region depicted and analyzed in Figure 2.

We thank the reviewer for the comment. The IHCs of apical cochlear region were analyzed in mice at 3 weeks of age. We have now added this to the figure legend.

(3) The discovery of orphan SGN neurites in RTN4RL2 KO mice is particularly intriguing. I wonder whether the additional Homer1-positive puncta illustrated in Figure 2 are present in these orphan SGN neurites, which would suggest that they may be functional. Conducting immunohistochemistry (IHC) labeling for type I SGN neurites using an anti-Tuj1 antibody, along with Homer1, would help localize the additional Homer1 puncta shown in Figure 2. Additionally, the "extra" Homer1 puncta appears less striking in the data presented in Figure 2-Supplement 2. Quantifying the number of Homer1 puncta in wild-type versus KO mice across different cochlear regions will help visualize the Figure 2-Supplement 2 data and relate the presence of extra neurites to the increased auditory brainstem response (ABR) thresholds observed at all frequencies.

We thank the reviewer for the comment and we agree that localizing orphan PSDs on the SGN neurites would be very useful. Unfortunately, the animal breeding license in the Göttingen lab had expired. At the time we received the reviews we only had access to 14-week-old animals and could not perform the stainings in animals which would have comparable age range to the rest of the study (3-4 weeks). The phenotype of extra Homer1 puncta was not as drastic in 14-week-old animals as it was in previously stained 3-week-old animals. Nevertheless, we still tried NF200, Homer1 and Vglut3 immunostainings in 14-week-old animals. We present representative single imaging planes of NF200, Homer1 and Vglut3 stainings in Author response image 4. Additionally, we provide exemplary images from 7-week-old RTN4RL2^-/-^, where it looks like that the orphan Homer1 puncta are found on calretinin positive neurites.

**Author response image 4. sa3fig4:** Attempts to localize “orphan” Homer1 patches on type I SGN neurites. (**A**) Single exemplary imaging planes of apical IHC region from RTN4RL2^+/+^ (left) and RTN4RL2^-/-^ (right) mice immunolabeled against NF200, Vglut3 and Homer1. White arrows show putative “orphan” Homer1 puncta on NF200 positive neurites. Scale bar = 5 mm. (**B**) Maximum intensity projections of representative confocal stacks of IHCs from RTN4RL2^-/-^ mice immunolabeled against Calretinin and Homer1. Scale bars = 5 mm. White arrows show possible “orphan” Homer1 puncta on Calretinin positive boutons.

(4) The authors noted a reduction in the number of GluA2/3-positive puncta in RTN4RL2 KOs, as shown in Figure 2-Supplement 3. However, in the Results section (page 5, line 124), it is unclear whether the authors refer to a reduction in fluorescence intensity or the number of puncta. Please clarify this.

We thank the reviewer for the comment. We refer to the number and have now added this to the manuscript.

(5) I find it particularly interesting that, despite the presence of smaller but synaptically engaged Homer1-positive SGN neurites, these appear to lack or present a reduction in the number of GluA2/3 puncta, and that GluA2/3 puncta are observed in non-ribbon juxtaposed neurites. Therefore, I suggest including GluA2/3 (Fig2 supplement 3) data in the main figure. It would be valuable to determine whether the orphan neurites express both Homer1 and GluA2/3, which could indicate that the defect is not solely due to reduced GluA2/3 expression at the formed synapses, but also to the presence of additional orphan synapses. I would also mention in the discussion how the phenotype of the RTN4L2 KO compares to the GluA2/3 KO and if the lack of GluA2/3 at the AZ could explain the increase in ABR threshold. Quantification of GluA2/3 puncta at the apical, middle, and basal region would also help understand the auditory phenotype of the KO mice.

We have changed Figure2-figure supplement 3 to become a main figure (Figure 3) based on the recommendation of the reviewer. We agree, that it would be valuable to perform immunohistochemistry combining anti-GluA2/3 and anti-Homer1 and anti-Ctbp2 antibodies to see if the “orphan” Homer1 patches house GluA2/3 not juxtaposing synaptic ribbons. Unfortunately, as mentioned above, due to the expiration of our animal breeding and experimentation licenses we did not manage to do those experiments. We have however performed stainings with anti-GluA4 antibodies and could not detect GluA4 signal in RTN4RL2^-/-^ mice (Figure 3-figure supplement 1). This potentially could explain the more drastic ABR threshold elevation in RTN4RL2^-/-^ mice compared to e.g. GluA3 KO mice. We have now made this clearer in our discussion.

(6) I suggest considering the use of color-blind friendly palettes for figures and graphs in this manuscript to enhance clarity and ensure that the findings are accessible to a wider audience and improve the overall effectiveness of the presentation. Please use color-blind-friendly schemes in Figure 1 and Figure 2 Supplement 3.

Done.

(7) Could you please explain what "XX {plus minus} Y, SD = W" means in the figure legends?

Mean ± SEM (standard error of the mean), SD (standard deviation) are indicated in the legends. In response to the reviewer comment we have now added an explanation in the Materials and Methods –> Data analysis and statistics section.

(8) Please include information about the ear tested (left or right or both).

Both ears were tested. Since there was no significant difference between right and left ear we did not further consider this factor. We will add this fact more precisely in the Material and methods section.

**Reviewer#3 (Recommendations for the authors):**
(1) Line 90: Why not show this control, it is a nice control.

Unfortunately, our recent attempts to perform RTN4RL2 immunostaining on cryosections were unsuccessful. Therefore, we decided to remove RTN4RL2 immunostaining from Figure 1 and have adjusted the results section accordingly.

(2) Line 94: Please provide a reference for these interactions.

Done.